# Make Autoencoder Great Again:
# Idempotent Autoencoder and Variational Idempotent Autoencoder

## Abstract

We propose a novel autoencoder paradigm, comprising the Idempotent Autoencoder (IAE) and the Variational Idempotent Autoencoder (VIAE), which integrates an idempotent loss to refine the latent space manifold. This approach enhances the continuity, consistency, and noise robustness of the learned manifold. Theoretically, we demonstrate that both models converge to the true data distribution, thereby improving latent space interpretability. Experimental results indicate that VIAE significantly outperforms standard VAEs in generation tasks and bridges the gap towards state-of-the-art one-stage generative autoencoders, achieving an FID of 11.74 on CIFAR-10. Furthermore, VIAE substantially improves latent space structure, exhibiting 50% higher Silhouette Coefficient and Calinski-Harabasz Index, and a 20% lower Davies-Bouldin Index compared to the vanilla VAE. In two-stage generation, VIAE demonstrates 14–33% faster convergence and superior generation quality compared to the autoencoders used in Latent Diffusion Models. Additionally, the IAE framework can theoretically leverage any pretrained encoder to construct a generative model for downstream tasks, a capability validated on toy datasets. The proposed models effectively address the trade-off between a structured latent space and high-quality generation, offering superior overall performance and robustness.

## 1. Introduction

In the pursuit of learning data distributions from a specified prior, generative modeling has established a diverse and robust landscape. Foundational frameworks such as Variational Autoencoders (VAEs) (Doersch, 2016; Daniel & Tamar, 2021; Dilokthanakul et al., 2016; Xu & Durrett, 2018; De Cao & Aziz, 2020; Huang et al., 2018), Generative Adversarial Networks (GANs) (Goodfellow et al., 2020; Zhu et al., 2017; Karras et al., 2019; Zhao et al., 2017), and Normalizing Flows (Rezende & Mohamed, 2015) have paved the way for modern synthesis. More recently, the field has witnessed a paradigm shift towards iterative models, where Diffusion Probabilistic Models (Ho et al., 2020; Song et al., 2020) and Flow Matching (Lipman et al., 2022; Holderrieth et al.) have achieved state-of-the-art synthesis quality. However, despite their success, these dominant paradigms face significant limitations. They are characterized by high computational costs due to multi-step sampling processes (Song et al., 2020; Ho et al., 2020; Lipman et al., 2022) and often fail to spontaneously learn a compact, semantically structured latent space (Lee et al., 2023; Wang et al., 2023), limiting both inference speed and interpretability.

Given these challenges, it is instructive to revisit the foundational autoencoder framework, which inherently offers efficient one-step generation and interpretable representations. Yet, VAEs historically struggle to match the visual fidelity of iterative methods. This limitation largely stems from the rigid Kullback-Leibler (KL) divergence constraint, which forces the aggregated posterior to match a simple, pre-defined prior (*e.g.*, a standard Gaussian). This forcing often results in the well-known "posterior hole" problem and over-smoothed generations, as the model prioritizes satisfying the prior over preserving the complex geometry of the data manifold.

To bridge this gap between efficient structure and high-fidelity synthesis, we re-examine the autoencoder through the lens of **Idempotence**. Our core insight is that an ideal autoencoder should act as a projection operator onto the data manifold: encoding a generated sample and decoding it *again* should yield the same result (consistency). By enforcing this idempotent property (*i.e.*, $f(f(x)) \approx f(x)$), we allow the model to learn the intrinsic geometry of the latent manifold rather than forcing it into a vacuous Gaussian hypersphere. This mechanism creates a "self-loop", enabling the model to accept noise from a simple prior and iteratively refine it into a valid latent vector through the

---

[1]Anonymous Institution, Anonymous City, Anonymous Region, Anonymous Country. Correspondence to: Anonymous Author <anon.email@domain.com>.

Preliminary work. Under review by the International Conference on Machine Learning (ICML). Do not distribute.

encoder-decoder cycle. In essence, this approach allows us to make the autoencoder **"great again and again"**: the idempotence ensures that repeated application of the model stabilizes the generation, transforming the autoencoder from a simple reconstructor into a robust, iterative generative projector.

We introduce a novel paradigm comprising the **Idempotent Autoencoder (IAE)** and the **Variational Idempotent Autoencoder (VIAE)**. Unlike traditional VAEs that passively penalize the distance to a prior, VIAE actively learns the manifold of the aggregated posterior via an idempotent loss inspired by Idempotent Generative Networks (IGN) (Shocher et al.). We extend IGN's theoretical guarantees from Euclidean space to the statistical manifold of probability measures, making it compatible with the probabilistic nature of VAEs. This formulation effectively pulls random noise onto the learned latent structure, addressing the critical trade-off between latent structure and generation quality.

Our main contributions are summarized as follows:

- We propose the IAE and VIAE frameworks, which integrate an idempotent optimization objective into the latent space to learn the true posterior manifold without the over-regularization of standard priors.

- We provide rigorous theoretical analysis extending operator-based idempotence to statistical manifolds, proving that VIAE converges to the true data distribution.

- We empirically demonstrate that VIAE significantly outperforms standard VAEs, achieving an FID of 11.74 on CIFAR-10 (compared to 82.70 for VAE) and exhibiting a 50% higher Silhouette Coefficient in latent clustering.

- We show that our framework can transform any pre-trained encoder into a generative model and serve as a superior latent compressor, improving the downstream diffusion models by 14-33%.

## 2. Method

This paper begins by presenting the background theory of the Idempotent Generative Network (IGN), a model trained to learn the data distribution $p_{data}(x)$ on manifold $\mathcal{M}$ from the source distribution $p(z)$ on sample manifold $\mathcal{Z}$. Subsequently, this paper will describe the core idea of integrating this type of manifold learning approach into an autoencoder architecture and provide an intuitive explanation of how this approach generates new samples. Finally, this paper will define the loss functions of Idempotent Autoencoder (IAE) and Variational Idempotent Autoencoder (VIAE), and then detail the training and sampling algorithms.

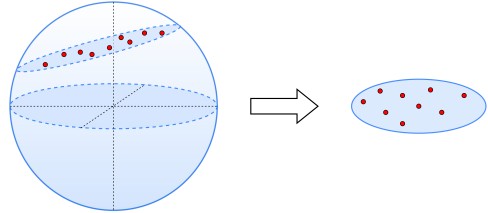

*Figure 1.* A picture shows a data manifold embedded in high-dimensional space.

### 2.1. Idempotent Generative Network

IGN learns a mapping from a sample manifold $\mathcal{Z}$ to the data manifold $\mathcal{M}$. The authors design the loss function based on the concept of fixed points. They treat the network $f_\theta$ as an idempotent operator (Shocher et al.). With the original concept, they train this network to ensure that the data manifold $\mathcal{M}$ is the unique set of fixed points for this operator and to make this operator a surjective mapping from the sampling manifold $\mathcal{Z}$ to the data manifold $\mathcal{M}$ simultaneously.

The loss function for the Idempotent Generative Network (IGN) is formulated in Eq. 1, where the three parts of this loss function are defined as shown in Eq. 2–4 respectively:

$$\mathcal{L}_{ign} = \mathcal{L}_{stable} - \lambda_t \cdot \mathcal{L}_{tight} + \lambda_i \cdot \mathcal{L}_{idem}, \quad (1)$$

$$\mathcal{L}_{stable} = \mathbb{E}_{x \sim p_{data}(x)} \big[ Div\big(f_\theta(x), x\big) \big], \quad (2)$$

$$\mathcal{L}_{tight} = \mathbb{E}_{z \sim p(z)} \big[ Div\big(f_\theta(f_{\theta^*}(z)), f_{\theta^*}(z)\big) \big], \quad (3)$$

$$\mathcal{L}_{idem} = \mathbb{E}_{z \sim p(z)} \big[ Div\big(f_{\theta^*}(f_\theta(z)), f_\theta(z)\big) \big]. \quad (4)$$

Here, $\theta^*$ represents a frozen copy of the current model $\theta$, $\lambda_i > 1$, and $D$ denotes a distance measure $d : \mathcal{X} \times \mathcal{X} \to \mathcal{R}$ that satisfies non-negativity and the identity of indiscernibles. With this loss function, IGN learns the data distribution $p_{data}(x)$ (more rigorously, $p_\theta(x) = p_{data}(x)$ a.e.) if and only if the loss meets its global minimum of 0, and all three parts equal 0 (The proof is shown in Section A.1).

Although IGN is built upon a strong theoretical foundation, achieving the theoretical limit in practice is difficult. As the original authors noted, IGN struggles with mode collapse and blurriness, similar to generative adversarial networks (Shocher et al.). This article posits that the reason for this limitation is that the original IGN attempts to learn the generative mapping directly in a high-dimensional data space. IGN tries to learn an idempotent operator that maps from a source distribution with full support in the high-dimensional space to a target distribution whose support is concentrated on a low-dimensional submanifold with zero Lebesgue measure in the high-dimensional space (a picture that makes this phenomenon easy to understand is shown in Figure 1); however, using a continuous function—namely, the neural network, to complete this task is hard, because

the optimal idempotent operator is likely to be discrete and too complex to approximate effectively.

## 2.2. Analysis on Manifold Learning in Autoencoder

Following this proposition, using IGN in the latent space to learn the latent data distribution could be significantly more effective than in the data space, analogous to the Latent Diffusion Model. However, this two-stage method is suboptimal; errors can accumulate across different parts of the model, and the whole model will occupy more storage space and lead to high latency while sampling.

However, IGN is very suitable for integration into the autoencoder architecture that naturally supplies a latent space within the model design, and becomes one model because the idempotent property in IGN is related to the cycle consistency in autoencoder. Holding this perspective, this article proposes two novel generative models: ***Idempotent AutoEncoder*** (IAE) and ***Variational Idempotent AutoEncoder*** (VIAE). Unlike other generative models based on autoencoder, IAE and VIAE no longer strictly restrict the aggregated posterior distribution to the prior distribution and turn to actively learning this posterior distribution through the idempotent property.

By regarding the Encoder-Decoder cycle $\mathcal{E} \circ \mathcal{D}(\cdot)$ as an IGN, IAE and VIAE can learn the posterior distribution constructed by the Encoder and obtain new posterior samples from any prior distribution. Next, with the well-trained Decoder, IAE and VIAE can recover data from the latent vector generated by the Encoder-Decoder cycle $\mathcal{E} \circ \mathcal{D}(\cdot)$ in the latent space.

## 2.3. Loss Functions and Sampling process

As shown in Section A.2, IAE has the ability to approximate the data distribution in the ideal case with the loss function shown in Eq. 5:

$$\mathcal{L}_{IAE} = \mathcal{L}_{rec} + \mathcal{L}_{ign} \tag{5}$$

$$
\begin{aligned}
&= \mathbb{E}_{x \sim p_{data}(x)} \big[ Div\big( \mathcal{D} \circ \mathcal{E}(x), x \big) \big] \\
&\quad + \mathbb{E}_{x \sim p_{data}(x)} \big[ Div\big( \mathcal{E} \circ \mathcal{D}(\mathcal{E}^*(x)), \mathcal{E}^*(x) \big) \big] \\
&\quad - \lambda_t \cdot \mathbb{E}_{z \sim p(z)} \big[ Div\big( \mathcal{E} \circ \mathcal{D}(\mathcal{E}^* \circ \mathcal{D}^*(z)), \mathcal{E}^* \circ \mathcal{D}^*(z) \big) \big] \\
&\quad + \lambda_i \cdot \mathbb{E}_{z \sim p(z)} \big[ Div\big( \mathcal{E}^* \circ \mathcal{D}^*(\mathcal{E} \circ \mathcal{D}(z)), \mathcal{E} \circ \mathcal{D}(z) \big) \big].
\end{aligned}
\tag{6}
$$

To generate new samples, firstly we need to obtain new samples of the aggregated posterior distribution $q_\varphi(z)$ from the prior distribution $p(z)$ with $\mathcal{E} \circ \mathcal{D}(\cdot)$, and then we use Decoder $\mathcal{D}(\cdot)$ construct the sampled latent vector into meaningful data point. Briefly, we can generate new data with $\mathcal{D} \circ \mathcal{E} \circ \mathcal{D}(\cdot)$ from the fixed prior distribution $p(z)$.

In the general Autoencoder, the data manifold is embedded

into a Euclidean latent space, so we can use IGN's convergence theory directly, while in Variational Autoencoder, the latent space is a Polish space composed of a series of random measures $q_\varphi(z|x)$. In order to use IGN in the probabilistic latent space, we firstly extend IGN's convergence theory to statistical manifolds, as shown in Section A.3. With this proof, we can use IGN to learn a statistical manifold rather than a data manifold, which is much more precise in the latent space of Variational Autoencoder. The loss function of VIAE is defined by

$$\mathcal{L}_{VIAE} = \mathcal{L}_{ELBO} + \mathcal{L}_{ign}. \tag{7}$$

We can prove that VIAE can converge to the data distribution, as shown in Section A.4.

Different from IAE, VIAE does not need to use the Encoder-Decoder cycle $\mathcal{E} \circ \mathcal{D}(\cdot)$ to obtain the aggregated posterior distribution $q_\varphi(x)$ in the ideal case, it can sample new data with Decoder from the prior distribution $p(z)$ directly, because the optimum of VIAE is a particular solution of VAE. While in practice, it is hard for VAE to maximize the Evidence Lower Bound, which means VAE is prone to generating low-quality samples. To overcome this challenge, VIAE can also use the Encoder-Decoder cycle $\mathcal{E} \circ \mathcal{D}(\cdot)$ to refine samples because the Encoder-Decoder cycle learns statistical manifolds composed of $q_\varphi(z|x)$ whose samples are used for the Decoder to reconstruct high-quality data.

## 3. Theoretical Result

This section is to derive useful latent-space properties implied by the **measure-theoretic idempotence** enforced by our loss.

### 3.1. Drift function: minimal requirements and identity property

**Definition 3.1** (Drift function). Let $(\mathcal{Y}, \mathcal{B}(\mathcal{Y}))$ be a measurable space and let $T$ be a measurable operator acting on $\mathcal{Y}$ (deterministic case) or acting on measures on $\mathcal{Y}$ (kernel case). A *drift function* is any measurable mapping $\delta : \mathcal{Y} \to [0, +\infty]$ that satisfies:

1. **Non-negativity:** $\delta(y) \geq 0$ for all $y$;

2. **Non-degeneracy:** $\sup_{y \in \mathcal{Y}} \delta(y) > 0$ (possibly $+\infty$).

3. **Identity of indiscernibles:** $Div(a, b) = 0 \iff a = b$,

**Definition 3.2** (Fixed-point set / learned manifold). Define the fixed-point set of $T$ by

$$\mathcal{S}_T := \{y \in \mathcal{Y} : \ \delta(y) = 0\} = \{y \in \mathcal{Y} : \ T(y) = y\}. \tag{8}$$

### 3.2. Latent self-loop as a kernel on the statistical manifold

For VIAE, the latent idempotent mechanism acts on the **statistical manifold** $\mathcal{P}(\mathcal{Z})$ (probability measures over latent space $\mathcal{Z}$). Define the decode-encode transition kernel $\mathcal{K}$ by

$$\mathcal{K}(z, A) := \int_{\mathcal{X}} q_\varphi(A \mid x) \, p_\theta(dx \mid z), \qquad (9)$$

where $A \in \mathcal{B}(\mathcal{Z})$. It can induce an operator $T$ on measures:

$$T(\pi)(A) := \int_{\mathcal{Z}} \mathcal{K}(z, A) \, d\pi(z), \quad \pi \in \mathcal{P}(\mathcal{Z}). \qquad (10)$$

### 3.3. Measure idempotence and stationarity

**Proposition 3.3** (Measure idempotence $\Rightarrow$ stationarity)**.** *Assume the idempotent term is minimized so that $T(T(\pi)) = T(\pi)$ holds for the generated family of measures (in particular for $\pi = p(z)$ or for the induced aggregated posterior). Then $\pi_{gen} := T(\pi)$ is a stationary measure:*

$$T(\pi_{gen}) = \pi_{gen}. \qquad (11)$$

**Corollary 3.4** (Support concentration on fixed points)**.** *If the stable term enforces $Div(\pi, T(\pi)) = 0$ for target posteriors, then every such posterior is supported on the fixed-point set:*

$$\pi(\mathcal{P}(\mathcal{Z}) \setminus \mathcal{S}_T) = 0, \quad equivalently \quad T(\pi) = \pi. \qquad (12)$$

*In particular, VIAE encourages $q_\varphi(\cdot \mid x)$ to be (approximately) stationary under the decode-encode kernel.*

### 3.4. Useful latent-space consequences

**Proposition 3.5** (Denoising / robustness with one-step projection)**.** *Suppose that $T(q_\varphi(\cdot \mid x)) \approx q_\varphi(\cdot \mid x)$ for $x \sim p_{data}$. Then a single decode-encode cycle acts as a projection back to the stationary posterior, removing the noise. Consequently, sampling through the latent can iteratively refine samples while preserving semantic content.*

**Proposition 3.6** (Support-set consistency (noise-invariant core))**.** *If $T(q_\varphi(\cdot \mid x)) = q_\varphi(\cdot \mid x)$, then for $z \sim q_\varphi(\cdot \mid x)$ and $\hat{x} \sim p_\theta(\cdot \mid z)$, we have $q_\varphi(\cdot \mid \hat{x}) = q_\varphi(\cdot \mid x)$ almost surely. Equivalently, the posterior support contracts to the noise-invariant core:*

$$\mathrm{supp}\big(q_\varphi(\cdot \mid x)\big) \subseteq \{z \in \mathcal{Z} : \; q_\varphi(\cdot \mid \hat{x}) = q_\varphi(\cdot \mid x)\}, \qquad (13)$$

*for $\hat{x} \sim p_\theta(\cdot \mid z)$.*

**Proposition 3.7** (Idealized segmentation and mutual singularity)**.** *In the deterministic decoder case, if each posterior concentrates on a decoder-fiber $\mathcal{D}^{-1}(\{x\})$, then for $x \neq x'$ the corresponding posteriors are mutually singular:*

$$q_\varphi(\cdot \mid x) \perp q_\varphi(\cdot \mid x'). \qquad (14)$$

*In particular, there exist disjoint measurable sets $A_x, A_{x'} \subset \mathcal{Z}$ such that $q_\varphi(A_x \mid x) = 1$ and $q_\varphi(A_{x'} \mid x') = 1$. This provides an explanation for the pixel segmentation and clustering about latent geometry observed in experiments.*

### 3.5. Fixed Encoder as a Generative Basis

From an information-theoretic perspective, if a pretrained encoder $E$ is sufficiently powerful to distinguish distinct data points, it effectively acts as a bijection between the data manifold $\mathcal{M}_x$ and the latent manifold $\mathcal{M}_z$.

Under this assumption, minimizing the IAE loss forces the composite function $\mathcal{E} \circ \mathcal{D}$ to align the distribution of generated latent vectors with the true aggregated posterior and $\mathcal{D}$ can generate new data with this learned distribution.

## 4. Experiments

In this section, we evaluate the proposed models, Idempotent Autoencoder (IAE) and Variational Idempotent Autoencoder (VIAE), regarding fundamental unconditional generative quality on toy datasets, efficacy in two-stage generative modeling, unconditional generation on standard benchmarks, and the learned latent geometric structures.

### 4.1. Toy Dataset

To intuitively understand the manifold learning capabilities of our frameworks, we conducted experiments on a 2D toy dataset consisting of the 2-moon mixtures, the 8 Gaussian mixtures, and the $5 \times 5$ grid mixtures. We compared our methods against the standard VAE and Idempotent Generative Network (IGN). The encoder and decoder for VAE, IAE, and VIAE consist of an MLP with a single 256-dimensional hidden layer, while the IGN utilizes an MLP with three 256-dimensional hidden layers.

**Generative Quality and Latent Space.** As illustrated in Figure 2a, the standard VAE suffers from the "posterior hole" and topology mismatch problems, resulting in blurry samples (Figure 2a, second column) and a densely packed, mismatched prior and approximate posterior in the latent space (Figure 2b, first column). While IGN generates sharp samples, it is prone to mode collapse as the number of modes increases, and it lacks a clustered latent representation that can be easily utilized in downstream tasks.

In contrast, both IAE and VIAE successfully learn the data manifold. VIAE, in particular, demonstrates the best generation quality, and IAE also exhibits the ability to learn the data distribution. We subsequently visualize the latent space as shown in Figure 2b. We observe that both IAE and VIAE form clusters. However, IAE cannot spontaneously form such regular geometric structures, whereas VIAE's latent space is highly structured. Unlike VAE, which rigidly

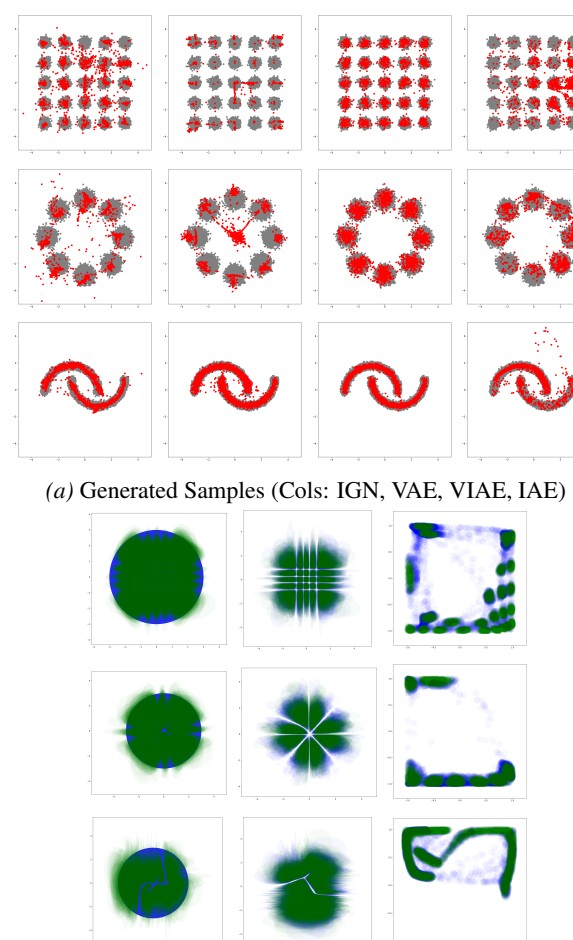

*(a)* Generated Samples (Cols: IGN, VAE, VIAE, IAE)

*(b)* Latent Space (Cols: VAE, VIAE, IAE)

*Figure 2.* Qualitative comparison on the Toy Dataset. (a) Generated samples in data space. (b) Visualization of the learned latent spaces.

forces the posterior to match a standard Gaussian $\mathcal{N}(0, I)$ (or other simple distributions), VIAE allows the aggregated posterior to adapt to the data topology while maintaining regularization via the KL divergence. This results in a latent space that is both compact and representative of the true data distribution; this spontaneous clustering simplifies the difficulty of learning the latent geometry in downstream tasks.

**Training a Generative Model with a Pretrained Encoder.** To verify our proposition mentioned in Theoretical Result 3.5, we test our framework on the 8 Gaussian mixtures and $5 \times 5$ grid mixtures datasets, which are more complex. Consistent with our theoretical analysis, the idempotent property allows us to turn a representation-learning model into a generative one. We validated this by post-training a decoder from scratch while keeping a pretrained encoder frozen (we trained this encoder with reconstruction loss to ensure that information can be sufficiently invertible on the

data support). As shown in Figure 3, the decoder learns to project random noise onto the data manifold defined by the fixed encoder. This confirms that our IAE loss function effectively forces the decoder to learn the generative projection operator. Consequently, this framework has the ability to turn any pre-trained autoencoder into a generative model without modifying its encoder, provided the encoder (excluding the output layer) is bijective.

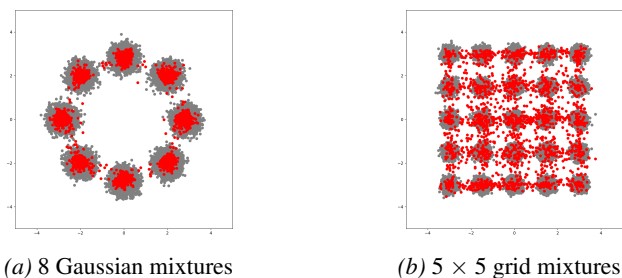

*(a)* 8 Gaussian mixtures      *(b)* $5 \times 5$ grid mixtures

*Figure 3.* Samples from a post-trained IAE Decoder with a frozen pretrained Encoder. The model successfully learns to project noise onto the data manifold.

### 4.2. Two-Stage Generation

We further evaluated the latent space modeling capabilities of VIAE in a two-stage generative framework, which is the foundation of modern Latent Diffusion Models (LDMs).

**Setup.** We conducted experiments on the **CelebA** dataset (Liu et al., 2015). To strictly compare the upper limit of the latent space structure, we adopted the architecture of the Latent Diffusion Model (Rombach et al., 2022) but with limited capacity. For the second-stage generative model, we trained a Flow Matching model (Lipman et al., 2022) with a DiT architecture (Peebles & Xie, 2023) on the latent representations extracted by different autoencoders (the architectures of all comparison models are identical). For the VAE baseline, we use the standard **AE+KL** loss used in Latent Diffusion Models (Rombach et al., 2022). For VQ-VAE, we follow the loss function used in VQ-VAE (Van Den Oord et al., 2017). For VIAE, we tested two different approaches for the posterior distribution: one trained to learn the posterior distribution, and the other trained to denoise the noisy posterior distribution.

**Results.** The results are illustrated in Figure 4. We measured the FID and FID-infinity across different numbers of function evaluations (NFEs) for Flow Matching and different numbers of idempotent cycles for VIAE. We find that the VIAE-based latent space significantly outperforms the AE+KL and VQ-VAE baselines. Specifically, VIAE decreases FID and FID-infinity by approximately **14% to 33%** (evaluated at 250 NFEs and 5 NFEs, respectively). Notably, the generation quality of VIAE converges within very few flow matching sampling steps. Compared to VAE, VIAE's

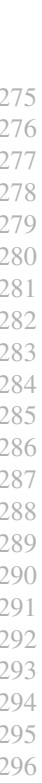

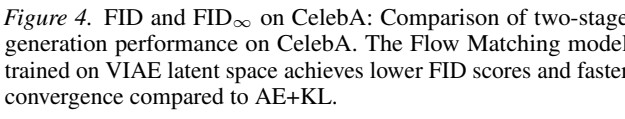

*Figure 4.* FID and FID$_\infty$ on CelebA: Comparison of two-stage generation performance on CelebA. The Flow Matching model trained on VIAE latent space achieves lower FID scores and faster convergence compared to AE+KL.

rapid convergence indicates that the learned aggregated posterior is much more accurate and exhibits robustness against noise in the latent space. This makes the probability flow trajectory straighter and easier for the diffusion/flow model to learn compared to the standard regularized latent space. Regarding VQ-VAE (Razavi et al., 2019), although it possesses a clustered structure, it is not well-organized, resulting in poor generation quality within limited model capacity; in contrast, VIAE possesses a better-organized and clustered structure, enabling it to outperform VQ-VAE.

### 4.3. Unconditional Generation

We evaluated the one-stage unconditional generation ability of our models on the **MNIST** (LeCun, 1998) and **CIFAR-10** (Krizhevsky et al., 2009) datasets. We compared our method against classic baselines including VAE, AAE, and GMVAE, as well as recent SOTA one-step approaches like Soft-IntroVAE and MeanFlow.

The quantitative results (FID-50k) are reported in Table 1. Standard VAEs struggle with these benchmarks (FID 47.95 on MNIST, 82.70 on CIFAR-10) due to the strict prior constraint causing blurry generation. VIAE achieves an FID of **2.20** on MNIST and **11.74** on CIFAR-10, significantly outperforming standard VAEs and approaching the performance of SOTA models. This demonstrates that by actively learning the aggregated posterior distribution instead of matching a fixed prior, VIAE can capture complex data distributions while retaining the efficiency of one-step sampling.

The gap in FID scores compared to Soft-IntroVAE (Daniel & Tamar, 2021) stems from our choice of the power spheri-

cal distribution as the posterior. While necessary for stable training and meaningful geometric structure, the overlap of power spherical distributions violates the theoretical requirement for mutually singular supports, leading to slight oversmoothing. In contrast, VIAE trades this pixel-level sharpness for a significantly more structured and interpretable latent space.

*Table 1.* FID(50k) scores of different models on MNIST and CIFAR-10 datasets. Lower is better.

| Model | MNIST | CIFAR-10 |
|---|---|---|
| AAE (Makhzani et al., 2015) | 4.31 | - |
| GMVAE (Dilokthanakul et al., 2016) | 36.00 | - |
| VAE (Doersch, 2016) | 47.95 | 82.70 |
| Soft-IntroVAE (Daniel & Tamar, 2021) | - | 4.60 |
| MeanFlow (Geng et al., 2025) | - | **2.92** |
| **IAE (Ours)** | 3.03 | 20.99 |
| **VIAE (Ours)** | **2.20** | 11.74 |

### 4.4. Unsupervised Latent Geometric Structure

The superior performance of VIAE in both one-step and two-stage generation tasks can be attributed to the unique geometric properties of its latent space. By revisiting the Toy Dataset (Section 4.1) and CelebA experiments, we can draw the following conclusions regarding the unsupervised clustering and structural properties:

- **Structure vs. Fragmentation:** As seen in Figure 2b, while IAE forms clusters, they can be fragmented due to the lack of regularization. VAE, conversely, over-regularizes, merging clusters indistinguishably to satisfy the standard Gaussian prior. VIAE strikes an optimal balance: the KL divergence introduces a strong structural regularization that pulls semantically similar points together, while the idempotent loss ensures these clusters remain distinct and sharp.

- **Benefit for Downstream Tasks:** This structured clustering is validated by the CelebA results. The fact that a Flow Matching model converges significantly faster on VIAE embeddings than on AE+KL embeddings suggests that the VIAE latent manifold is continuous and "smooth" in a way that aligns with the data semantics, rather than being artificially forced into a hyper-sphere. This geometric alignment makes VIAE a superior candidate for latent-space generative modeling compared to traditional VQ-VAEs (which are discrete) or standard AE+KL (which is not well-structured).

To quantitatively verify the structural advantages of VIAE, we analyzed the latent geometry on both simple (MNIST) and complex (CIFAR-10) datasets.

**The Idempotent Projection Effect on CIFAR-10.** We evaluated the intrinsic clustering quality of the latent space using label-free metrics: Silhouette Coefficient, Calinski-Harabasz (CH) Index, and Davies-Bouldin (DB) Index. We compared the standard encoding $z = E(x)$ and the "idempotent projection" $z^* = E(D(z))$ for both VAE and VIAE.

As shown in Figure 5, we observe that: **Iterative Refinement:** Standard VAEs (orange lines) show no change between the initial encoding and the re-encoded projection. This confirms that VAEs do not inherently learn a self-refining manifold. In contrast, VIAE (blue lines) exhibits a dramatic improvement in all metrics after the idempotent projection (higher Silhouette Coefficient, CH index, and lower DB index). This empirically validates our theoretical proposition that VIAE's encoder learns a vector field where data points flow towards the noise-invariant core (acting as fixed points in Euclidean space), effectively filtering out noise and sharpening semantic boundaries.

**Continuity vs. Discreteness:** Interestingly, the *initial* one-step encoding of VIAE (solid blue lines) performs similarly to the standard VAE. This is a desirable property, indicating that the KL divergence successfully regularizes the global topology to be continuous and smooth, preventing the scattered latent space observed in autoencoders and VQ-VAE. However, the idempotent loop allows VIAE to dynamically contract this smooth space into sharp, distinct clusters when needed.

**Clustering Accuracy on MNIST.** On the MNIST dataset, we observed that VIAE improves the unsupervised K-Means clustering accuracy from 56% (VAE baseline) to **64%**.

**Discussion on Evaluation Metrics.** We employ different evaluation strategies for the two datasets based on their manifold nature. MNIST represents a "geometric manifold" where pixel-space similarity (digit shape) strongly correlates with semantic labels; thus, K-Means accuracy is a direct proxy for generation quality. Conversely, CIFAR-10 represents a complex "semantic manifold" where visual modes do not strictly align with Euclidean distance. Therefore, for CIFAR-10, the significant boost in internal clustering metrics (Figure 5) is a more robust indicator that VIAE successfully groups similar visual patterns into tight, but well-separated parts without supervision. We argue that in CIFAR-10, aligning strictly with human-annotated labels are a limited proxy for assessing a generative model's latent geometry. Human labels often conflate visually distinct modes (e.g., a 'plane' on the ground vs. in the sky) into a single class. Our results show that VIAE achieves high internal clustering metrics, indicating a spontaneous emergence of structured manifolds. This suggests that VIAE captures the intrinsic visual modalities of the data, separating patterns based on their inherent statistical properties rather than being forced into predetermined semantic buckets. This

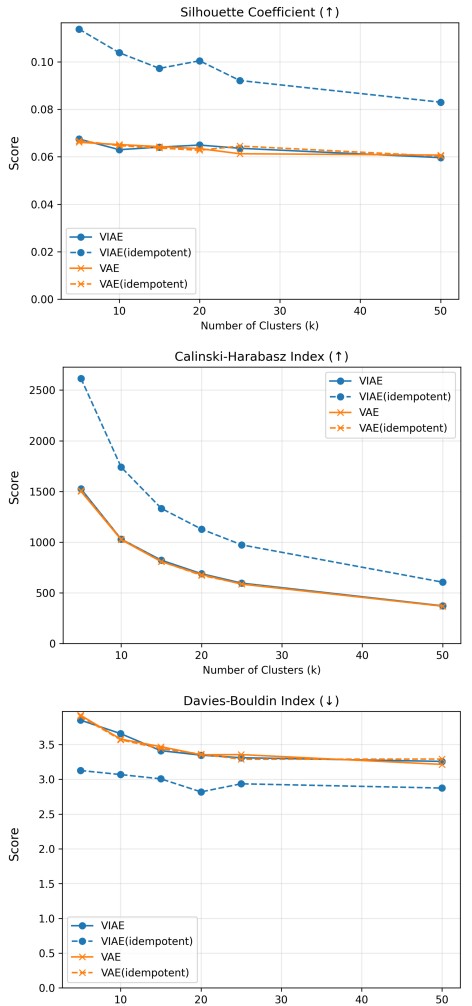

*Figure 5.* Geometric Structure Analysis on CIFAR-10. The **solid lines** represent the initial latent code $z \sim E(x)$, while the **dashed lines** represent the projected code $z^* = E(D(z))$. VIAE shows significant structural improvement after self-looping, whereas VAE remains unchanged.

capability to discover "natural clusters" without supervision is a key contribution of our measure-theoretic idempotent framework.

## 5. Related Works

**Adversarial Autoencoders and Variational Bayes (Makhzani et al., 2015; Mescheder et al., 2017)** Adversarial Autoencoders (AAE) and Adversarial Variational Bayes (AVB) enhance generative modeling by adversarially matching the aggregated posterior $q_\varphi(z)$ to a fixed prior $p(z)$. Unlike these methods, which *constrain* $q_\varphi(z)$ toward a simple reference, IAE/VIAE *avoid hard prior-matching*. Instead, they learn the aggregation distribution via an idempotent latent operator (the decode-encode

self-loop), effectively projecting noise onto the learned latent manifold.

**Vector-Quantized and Segmented Autoencoders (Razavi et al., 2019)** VQ-VAE utilizes a discrete bottleneck to achieve segmented representations and high perceptual quality. While VQ-VAE enforces segmentation via hard assignment to a codebook, VIAE induces an analogous effect through *idempotence*. The posterior $q_\varphi(z \mid x)$ becomes invariant under the decode-encode operator, producing partitioned latent regions within a continuous space.

**Disentanglement (Burgess et al., 2018; Kim & Mnih, 2018)** Methods like $\beta$-VAE and FactorVAE promote factorial latent dimensions by penalizing total correlation. Our framework is compatible with these objectives; the idempotent loss, which enforces posterior invariance under a latent projector, can be combined with TC-based regularizers to further enhance factorization.

**Two-Stage Latent Generative Models (Rombach et al., 2022)** Two-stage pipelines (e.g., Latent Diffusion, Flow Matching) first compress data via an autoencoder, then train a generative model in the latent space. While offering high synthesis quality, this increases training and inference complexity. IAE/VIAE integrate manifold learning directly into the autoencoder, establishing a structured latent manifold in a single stage to ensure interpretability and efficient sampling.

**IntroVAE and Soft-IntroVAE (Huang et al., 2018; Daniel & Tamar, 2021)** IntroVAE-based methods employ a self-adversarial mechanism where the autoencoder acts as a discriminator. VIAE shares this self-adversarial nature (via stability and idempotence) but distinguishes itself by enforcing a **measure-idempotent** operator. This leads to unique structural properties—posterior stationarity and support contraction—that facilitate unsupervised clustering. While adversarial VAEs often prioritize lower FIDs at the cost of latent topology, VIAE balances perceptual quality with a significantly more structured and interpretable latent manifold.

**Idempotent Generative Network (IGN) (Shocher et al.)** IGN trains an idempotent operator to project samples onto the data manifold in pixel space, but suffers from optimization challenges like mode collapse. We relocate the idempotent principle to the compact latent space, aligning it with the natural encoder-decoder cycle. This yields a unified, one-stage generative model with both theoretical guarantees and enhanced stability.

## 6. Limitations and Future Work

While VIAE establishes a rigorous framework for learning latent manifolds via idempotence, our analysis and exper-

iments reveal several limitations that are open for future research.

**Gaussian Assumption vs. Mutual Singularity.** According to our theory (Appendix A.4), perfect idempotence requires that the probability distributions $q_\phi(z|x)$ for different data points should be mutually singular. However, in practice, we model the posterior distribution as Gaussian or Uniform distributions for easier computation and training stability. These distributions naturally leak into each others' support, breaking the ideal separation rule. This creates a compromise: VIAE achieves a well-organized latent space where similar data cluster neatly, although the Gaussian approximation smooths the manifold. This prevents VIAE from matching the SOTA FID scores of adversarial methods like Soft-IntroVAE. Future improvements could employ distributions with strict boundaries (like Truncated Normal) or flexible non-parametric methods to better adhere to the theory while maintaining smooth optimization.

**Generalizing to Operator Generative Networks (OGN).** Currently, IGN, IAE, and VIAE learn an *idempotent projector* ($T \circ T = T$), which represents a specific instance of a broader class of models we term **Operator Generative Networks (OGN)**. These models and their proofs (Section A.1, Section A.2, and Section A.4) rely on a strict definition of the drift function $\delta$, specifically requiring the *identity of indiscernibles* (i.e., $\delta(z) = 0 \iff T(z) = z$) to enforce element-wise stability.

A promising direction for future work is to **relax the requirements on** $\delta$ within the Relaxing Idempotent Generative Networks framework, as shown in Section A.3. By removing the strict identity constraint, we can extend the framework to learn **aggregated measure manifold preserving operators** where $T_{\#}p_{data} = p_{data}$ but $T(z) \neq z$. This generalizes the generative process from static projection to dynamic traversal, allowing the model to learn transformations or vector fields on the manifold. Such an OGN would enable controlled semantic matching, effectively unifying idempotent networks, flow-based models, and Markov Chain Monte Carlo methods under a single operator-theoretic paradigm.

## Impact Statement

This paper presents work whose goal is to advance the field of Machine Learning. There are many potential societal consequences of our work, none which we feel must be specifically highlighted here.

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

## A. Proofs

### A.1. Idempotent Generative Network in Euclidean space

**Setting.** Let $\mathcal{X} \subseteq \mathbb{R}^n$ be the ambient space. Let $P_x$ be the data distribution and $P_z$ be a source distribution (e.g., Gaussian) on $\mathcal{X}$. Let $f_\theta : \mathcal{X} \to \mathcal{X}$ be a measurable map. Define the *drift*

$$\delta_\theta(y) \triangleq Div(y, f_\theta(y)) \in [0, M_\theta], \tag{15}$$

where $Div$ is a nonnegative distance-like function and we assume $M_\theta \triangleq \sup_y Div(y, f_\theta(y)) > 0$.

Let $P_\theta$ denote the distribution of $y = f_\theta(z)$ when $z \sim P_z$.

**IGN idealized objective (two-task form).** Define the reconstruction-tightness objective as

$$L_{\mathrm{rt}}(\theta; \theta^\star) \triangleq E_{x \sim P_x}\big[\delta_\theta(x)\big] - \lambda_t \, E_{z \sim P_z}\big[\delta_\theta(f_{\theta^\star}(z))\big], \qquad \lambda_t \in (0, 1], \tag{16}$$

where $\theta^\star$ is a frozen copy of $\theta$ of the current networks. Define the idempotence objective (optimized through the inner instantiation)

$$L_{\mathrm{idem}}(\theta; \theta^\star) \triangleq E_{z \sim P_z}\big[\delta_{\theta^\star}(f_\theta(z))\big]. \tag{17}$$

**Assumption A.1** (Ideal capacity and global optimality). There exists $\theta^\star$ such that $\theta^\star$ is simultaneously a global minimizer of $L_{\mathrm{rt}}(\theta; \theta^\star)$ and $L_{\mathrm{idem}}(\theta; \theta^\star)$ over all measurable maps $f_\theta$.

**Theorem A.2** (IGN convergence under ideal conditions). *Under Assumption A.1: (i) if $\lambda_t = 1$, then necessarily $P_{\theta^\star} = P_x$; (ii) if $\lambda_t \in (0, 1)$, then $d_{\mathrm{TV}}(P_{\theta^\star}, P_x) \leq \frac{1 - \lambda_t}{\lambda_t}$ and hence $d_{\mathrm{TV}}(P_{\theta^\star}, P_x) \to 0$ as $\lambda_t \uparrow 1$.*

*Proof.* **Step 1: Pointwise characterization of the optimal drift for $L_{\mathrm{rt}}$.**

We write $p_x$ and $p_{\theta^\star}$ for densities of $P_x$ and $P_{\theta^\star}$. Change variables $y := x$ in the first expectation and $y := f_{\theta^\star}(z)$ in the second, yielding

$$L_{\mathrm{rt}}(\theta; \theta^\star) = \int_\mathcal{X} \delta_\theta(y)\,\big(p_x(y) - \lambda_t p_{\theta^\star}(y)\big)\, dy. \tag{18}$$

Since $0 \leq \delta_\theta(y) \leq M$ and the integrand separates pointwise in $y$, the minimum over $\delta_\theta(\cdot)$ is attained by

$$\delta_\theta^{\mathrm{rt},\star}(y) = \begin{cases} 0, & p_x(y) \geq \lambda_t p_{\theta^\star}(y), \\ M_\theta, & p_x(y) < \lambda_t p_{\theta^\star}(y). \end{cases} \tag{19}$$

Indeed, if $p_x(y) - \lambda_t p_{\theta^\star}(y) \geq 0$, choosing $\delta_\theta(y) = 0$ minimizes the contribution; otherwise the coefficient is negative and choosing $\delta_\theta(y) = M_\theta$ minimizes the integral.

**Step 2: $L_{\mathrm{idem}}$ becomes an indicator probability.**

By definition,

$$L_{\mathrm{idem}}(\theta; \theta^\star) = E_{z \sim P_z}\big[\delta_{\theta^\star}(f_\theta(z))\big]. \tag{20}$$

Plugging in the optimal drift form (19) for $\delta_{\theta^\star}$ gives

$$L_{\mathrm{idem}}(\theta; \theta^\star) = M_\theta \cdot E_{z \sim P_z}\Big[1_{\{p_x(y) - \lambda_t p_{\theta^\star}(y) < 0\}}\Big], \tag{21}$$

since $y = f_\theta(z)$ is distributed as $P_\theta$.

**Step 3: Identify the global minimizer distribution.**

If for any measurable $A \subseteq \mathcal{X}$ yields

$$P_x(A) = P_{\theta^\star}(A) \qquad . \tag{22}$$

The loss will be 0.

Meanwhile, the loss in (21) is always nonnegative, hence its global minimum is 0.

Therefore any global minimizer $\theta^\star$ must satisfy

$$M_\theta \cdot E_{z \sim P_z}\Big[1_{\{p_x(y) - \lambda_t p_{\theta^\star}(y)\}}\Big] = 0 \quad \implies \quad p_x(y) \geq \lambda_t p_{\theta^\star}(y) \text{ for } P_{\theta^\star} \text{ a.e. } y. \tag{23}$$

Integrating (23) over any measurable $A \subseteq \mathcal{X}$ yields

$$P_x(A) \geq \lambda_t P_{\theta^\star}(A) \qquad \forall A. \tag{24}$$

If $\lambda_t = 1$, then (24) implies $P_x(A) \geq P_{\theta^\star}(A)$ for all $A$. Since both are probability measures, equality of total mass forces $P_x(A) = P_{\theta^\star}(A)$ for all $A$, hence $P_{\theta^\star} = P_x$.

If $\lambda_t \in (0, 1)$, define the Radon-Nikodym derivative $r = \frac{dP_{\theta^\star}}{dP_x}$. From (24) we obtain $r \leq 1/\lambda_t$ $P_x$ a.e. Then the total variation distance satisfies

$$d_{\mathrm{TV}}(P_{\theta^\star}, P_x) = \frac{1}{2}\int_{\mathcal{X}} |r - 1|\, dP_x = \int_{\mathcal{X}} (r - 1)_+\, dP_x \leq \Big(\frac{1}{\lambda_t} - 1\Big)\int_{\mathcal{X}} dP_x = \frac{1 - \lambda_t}{\lambda_t}. \tag{25}$$

Letting $\lambda_t \uparrow 1$ proves the claim. $\qquad\qquad\square$

### A.2. Convergence Theory of Idempotent Autoencoder

**IAE loss.** Recall the IAE objective in Eq. (5):

$$\mathcal{L}_{IAE} = \mathbb{E}_{x \sim p_{data}(x)}\Big[Div\big(\mathcal{D} \circ \mathcal{E}(x), x\big)\Big] \tag{26}$$

$$+ \mathbb{E}_{x \sim p_{data}(x)}\Big[Div\big(\mathcal{E} \circ \mathcal{D}(\mathcal{E}^*(x)), \mathcal{E}^*(x)\big)\Big] \tag{27}$$

$$- \lambda_t \cdot \mathbb{E}_{z \sim p(z)}\Big[Div\big(\mathcal{E} \circ \mathcal{D}(\mathcal{E}^* \circ \mathcal{D}^*(z)), \mathcal{E}^* \circ \mathcal{D}^*(z)\big)\Big] \tag{28}$$

$$+ \lambda_i \cdot \mathbb{E}_{z \sim p(z)}\Big[Div\big(\mathcal{E}^* \circ \mathcal{D}^*(\mathcal{E} \circ \mathcal{D}(z)), \mathcal{E} \circ \mathcal{D}(z)\big)\Big]. \tag{29}$$

Throughout this proof we require $Div$ with identity of indiscernibles and has a non-negative supremum.

**Realizability for $\mathcal{L}_{IAE} = 0$ with infinite capacity.** With infinity capacity of measurable functions, there exists a measurable decoder $\mathcal{D}_{opt} : \mathcal{Z} \to \mathcal{X}$ such that

$$(\mathcal{D}_{opt})_\sharp p(z) = p_{data}(x), \tag{30}$$

and letting $\mathcal{X}_{opt} := \mathrm{Im}(\mathcal{D}_{opt})$, with Kuratowski-Ryll-Nardzewski Selection Theorem, there exists a measurable right-inverse $\mathcal{E}_{opt} : \mathcal{X}_{opt} \to \mathcal{Z}$ satisfying

$$\mathcal{D}_{opt}(\mathcal{E}_{opt}(x)) = x, \qquad \forall x \in \mathcal{X}_{opt}, \tag{31}$$

and the data distribution is supported on $\mathcal{X}_{opt}$:

$$p_{data}(\mathcal{X} \setminus \mathcal{X}_{opt}) = 0. \tag{32}$$

**Claim.** Under the above assumption, there exists $(\mathcal{E}, \mathcal{D})$ such that every term in Eq. (5) equals 0, hence $\mathcal{L}_{IAE} = 0$ is achievable.

**Proof.** Take $\mathcal{D} = \mathcal{D}_{opt}$ and $\mathcal{E} = \mathcal{E}_{opt}$ on $\mathcal{X}_{opt}$ and extend $\mathcal{E}$ arbitrarily to all of $\mathcal{X}$. As IGN, $\mathcal{E}^\star$ and $\mathcal{D}^\star$ are the frozen copies of the current networks, i.e., $\mathcal{E}^\star = \mathcal{E}$ and $\mathcal{D}^\star = \mathcal{D}$. Define $g := \mathcal{E} \circ \mathcal{D} : \mathcal{Z} \to \mathcal{Z}$.

**(1) Reconstruction term.** For $x \sim p_{data}$, we have $x \in \mathcal{X}_{opt}$ almost surely. Hence

$$\mathcal{D} \circ \mathcal{E}(x) = \mathcal{D}_{opt}(\mathcal{E}_{opt}(x)) = x \tag{33}$$

Thus the first term in Eq. (5) equals $\mathbb{E}_{x \sim p_{data}}[Div(x, x)] = 0$ by $Div(u, u) = 0$.

**(2) Stable term in latent IGN.** Since $\mathcal{E}^\star = \mathcal{E}$, the stable term is

$$\mathbb{E}_{x \sim p_{data}} \Big[ Div\big(g(\mathcal{E}(x)),\ \mathcal{E}(x)\big) \Big] = \mathbb{E}_{x \sim p_{data}} \Big[ Div\big(\mathcal{E}(\mathcal{D}(\mathcal{E}(x))),\ \mathcal{E}(x)\big) \Big]. \tag{34}$$

But for $x \sim p_{data}$, we already have $\mathcal{D}(\mathcal{E}(x)) = x$, hence $\mathcal{E}(\mathcal{D}(\mathcal{E}(x))) = \mathcal{E}(x)$, and therefore the integrand is $Div(u, u) = 0$ a.e. So the stable term is 0.

**(3) Tightness term in latent IGN.** With $\mathcal{E}^\star = \mathcal{E}$ and $\mathcal{D}^\star = \mathcal{D}$, the tightness divergence is

$$Div\big(g(g(z)),\ g(z)\big), \qquad z \sim p(z), \tag{35}$$

since $g(g^\star(z)) = g(g(z))$ and $g^\star(z) = g(z)$. Now note that for any $z \in \mathcal{Z}$, $\mathcal{D}(z) \in \mathcal{X}_{opt}$ by definition of image. Therefore

$$\mathcal{D}(\mathcal{E}(\mathcal{D}(z))) = \mathcal{D}(z) \quad \Longrightarrow \quad g(g(z)) = \mathcal{E}(\mathcal{D}(\mathcal{E}(\mathcal{D}(z)))) = \mathcal{E}(\mathcal{D}(z)) = g(z), \tag{36}$$

where we used $\mathcal{D} \circ \mathcal{E} = \text{Id}$ on all of $\mathcal{X}_{opt}$ (not merely on the data support). Hence $Div(g(g(z)), g(z)) = Div(u, u) = 0$ for $z \sim p(z)$, so the tightness term equals 0.

**(4) Idempotence term in latent IGN.** Similarly, with starred networks equal to the current ones, the idempotence divergence is again

$$Div\big(g(g(z)),\ g(z)\big), \tag{37}$$

which we have shown is 0 for $p(z)$ a.e.. Hence the idempotence term is 0.

Combining (1)-(4), each expectation in Eq. (5) is 0, and thus $\mathcal{L}_{IAE} = 0$ is achievable under the stated realizability assumption. $\qquad\square$

**Latent IGN reparameterization.** Define the latent-space maps

$$g_\theta \triangleq \mathcal{E} \circ \mathcal{D} : \mathcal{Z} \to \mathcal{Z}, \qquad g_{\theta^\star} \triangleq \mathcal{E}^\star \circ \mathcal{D}^\star : \mathcal{Z} \to \mathcal{Z}. \tag{38}$$

Then the three latent terms (27)-(29) are exactly the IGN stable/tightness/idempotence objectives on $\mathcal{Z}$ with generator $g_\theta$ and frozen model $g_{\theta^\star}$:

$$L_{\text{rt}}^{(Z)}(\theta; \theta^\star) \triangleq \mathbb{E}_{x \sim p_{data}} \Big[ Div\big(g_\theta(\mathcal{E}^\star(x)),\ \mathcal{E}^\star(x)\big) \Big] - \lambda_t \, \mathbb{E}_{z \sim p(z)} \Big[ Div\big(g_\theta(g_{\theta^\star}(z)),\ g_{\theta^\star}(z)\big) \Big], \tag{39}$$

$$L_{\text{idem}}^{(Z)}(\theta; \theta^\star) \triangleq \mathbb{E}_{z \sim p(z)} \Big[ Div\big(g_{\theta^\star}(g_\theta(z)),\ g_\theta(z)\big) \Big], \tag{40}$$

so that the latent IGN part of $\mathcal{L}_{IAE}$ is $L_{\text{rt}}^{(Z)} + \lambda_i L_{\text{idem}}^{(Z)}$ (with $\lambda_t \in (0, 1]$ the same coefficient as in IGN).

**Ideal condition.** We adopt the same ideal assumption as in IGN: there exists a frozen model $\theta^\star$ such that $\theta^\star$ is simultaneously a global minimizer of $L_{\text{rt}}^{(Z)}(\theta; \theta^\star)$ and $L_{\text{idem}}^{(Z)}(\theta; \theta^\star)$ over all measurable maps.

**Theorem A.3** (Latent distribution matching for IAE). *Under the ideal condition above: if $\lambda_t = 1$, then the distribution of $g_{\theta^\star}(Z)$ equals the distribution of $\mathcal{E}^\star(X)$*

*Proof.* This is a direct application of the Euclidean space IGN theorem (Appendix A.1) on the latent space $\mathcal{Z}$, with target distribution $q_\varphi(z) = \int_\mathcal{X} q_\varphi(z|x) dP_{data}(x)$ and drift $\delta(u) = Div(u, g_\theta(u))$ bounded in $[0, M_\theta]$ with $\sup \delta > 0$. The optimization tasks coincide with $L_{\text{rt}}^{(Z)}$ and $L_{\text{idem}}^{(Z)}$ by construction. $\qquad\square$

**Theorem A.4** (Generative correctness of IAE). *Assume $\mathcal{L}_{IAE}$ reaches its global minimum and $\lambda_t = 1$ (so that the latent matching in Theorem A.3 holds and the reconstruction term (26) attains 0), then the IAE sampler*

$$G(z) \triangleq \mathcal{D}\big(g_{\theta^\star}(z)\big) = \mathcal{D}\big(\mathcal{E} \circ \mathcal{D}(z)\big) = \mathcal{D} \circ \mathcal{E} \circ \mathcal{D}(z) \tag{41}$$

*satisfies $G(Z) \sim p_{data}$ when $Z \sim p(z)$.*

*Proof.* Let $X \sim p_{data}$ and define $U := \mathcal{E}^\star(X)$. By Theorem A.3 with $\lambda_t = 1$, we have $g_{\theta^\star}(Z) \overset{d}{=} U$. Therefore $\mathcal{D}(g_{\theta^\star}(Z)) \overset{d}{=} \mathcal{D}(U)$. If the reconstruction term is 0 (and $Div$ has identity of indiscernibles), then $\mathcal{D}(\mathcal{E}^\star(X)) = X$ holds $p_{data}$ a.e., hence $\mathcal{D}(U) \sim p_{data}$. Thus $G(Z) \sim p_{data}$. (If one does not assume identity of indiscernibles, the same conclusion follows whenever $Div(\mathcal{D}(\mathcal{E}^\star(X)), X) = 0$ implies equality in distribution of $\mathcal{D}(\mathcal{E}^\star(X))$ and $X$.) $\qquad\square$

### A.3. Relaxing Idempotent Generative Network in Polish space

**Measure-theoretic formulation.** Let $\mathcal{X}$ be a Polish space with Borel $\sigma$-algebra $\mathcal{B}(\mathcal{X})$. Let $\mu$ be the target (data) probability measure and $\nu$ be the generated probability measure. Let $\lambda \in (0, 1]$.

A *drift* is any measurable $\delta : \mathcal{X} \to [0, M]$ for some $M \in (0, \infty]$ where $\delta$ is a nonnegative function and we assume $M \triangleq \sup_x \delta(x) > 0$ (Here, $\delta$ could violate identity of indiscernibles).

Define the reconstruction-tightness functional

$$\mathcal{L}_{\mathrm{rt}}(\delta; \nu) \triangleq \int_{\mathcal{X}} \delta \, d\mu - \lambda \int_{\mathcal{X}} \delta \, d\nu = \int_{\mathcal{X}} \delta \, d(\mu - \lambda\nu), \tag{42}$$

and the idempotence functional (projection step)

$$\mathcal{L}_{\mathrm{idem}}(\nu; \delta) \triangleq \int_{\mathcal{X}} \delta \, d\nu. \tag{43}$$

**Assumption A.5** (Idealized alternating global optimality). There exists a pair $(\nu^\star, \delta^\star)$ such that:

1. $\delta^\star$ minimizes $\mathcal{L}_{\mathrm{rt}}(\delta; \nu^\star)$ over all measurable $\delta : \mathcal{X} \to [0, M]$;

2. $\nu^\star$ minimizes $\mathcal{L}_{\mathrm{idem}}(\nu; \delta^\star)$ over all probability measures on $\mathcal{X}$.

**Theorem A.6** (TV convergence in Polish space). *Under Assumption A.5, the fixed-point generated measure $\nu^\star$ satisfies*

$$d_{\mathrm{TV}}(\nu^\star, \mu) \leq \frac{1 - \lambda}{\lambda}, \tag{44}$$

*and hence $d_{\mathrm{TV}}(\nu^\star, \mu) \to 0$ as $\lambda \uparrow 1$. If $\lambda = 1$, then necessarily $\nu^\star = \mu$.*

*Proof.* **Step 1: Optimal drift via Hahn decomposition (bounded case).** Consider the signed measure $\sigma \triangleq \mu - \lambda\nu^\star$. By the Hahn decomposition theorem, there exist disjoint measurable sets $(P, N)$ with $P \cup N = \mathcal{X}$ such that $\sigma \geq 0$ on $P$ and $\sigma \leq 0$ on $N$. For any $\delta \in [0, M_\theta]$,

$$\int_{\mathcal{X}} \delta \, d\sigma = \int_P \delta \, d\sigma + \int_N \delta \, d\sigma \tag{45}$$

and the lower bound is attained by choosing $\delta^\star = 0$ on $P$ and $\delta^\star = M$ on $N$.

Thus, an optimizer can be taken as

$$\delta^\star(x) = M \cdot \mathbf{1}_N(x), \tag{46}$$

.

**Step 2: The idempotence step forces $\nu^\star(N) = 0$.** By (46),

$$\mathcal{L}_{\mathrm{idem}}(\nu; \delta^\star) = \int_{\mathcal{X}} \delta^\star \, d\nu = M \cdot \nu(N). \tag{47}$$

The minimum over all probability measures $\nu$ is 0, achieved iff $\nu(N) = 0$. By Assumption A.5(2), we conclude $\nu^\star(N) = 0$.

Since $N$ is a *negative* set for $\sigma = \mu - \lambda\nu^\star$, we have $\sigma(N) \leq 0$, i.e., $\mu(N) - \lambda\nu^\star(N) \leq 0$. Using $\nu^\star(N) = 0$ yields $\mu(N) \leq 0$, hence $\mu(N) = 0$.

**Step 3: Domination and TV bound.** We have shown $\mu(N) = \nu^\star(N) = 0$, so for any measurable $A$,

$$\sigma(A) = \sigma(A \cap P) + \sigma(A \cap N) = \sigma(A \cap P) \geq 0, \tag{48}$$

i.e. $\mu(A) \geq \lambda\nu^\star(A)$ for all $A$. Therefore $\nu^\star \ll \mu$ and the Radon-Nikodym derivative satisfies $\frac{d\nu^\star}{d\mu} \leq \frac{1}{\lambda} \mu$ a.e., and the total variance has a upper bound,

$$d_{\mathrm{TV}}(\nu^\star, \mu) = \int_{\mathcal{X}} \left( \frac{d\nu^\star}{d\mu} - 1 \right)_+ d\mu \leq \left( \frac{1}{\lambda} - 1 \right) \int_{\mathcal{X}} d\mu = \frac{1 - \lambda}{\lambda}. \tag{49}$$

If $\lambda = 1$, domination gives $\nu^\star = \mu$. $\qquad\square$

### A.4. Convergence Theory of Variational Idempotent Autoencoder

**Claim (VIAE optimum is a VAE special optimum).** Let

$$\mathcal{L}_{VIAE}(\theta, \varphi) = \mathcal{L}_{VAE}(\theta, \varphi) + \mathcal{L}_{ign\_sm}(\theta, \varphi), \qquad \mathcal{L}_{ign\_sm} \geq 0. \tag{50}$$

We show that there exists a parameter pair $(\theta, \varphi)$ such that $\mathcal{L}_{VAE}$ attains its global minimum (ELBO is maximal) and simultaneously $\mathcal{L}_{ign\_sm} = 0$. Consequently, any global minimizer of $\mathcal{L}_{VIAE}$ must also minimize $\mathcal{L}_{VAE}$, hence $p_\theta(x) = p_{data}(x)$ at the VIAE optimum and the model is generative.

**Minimal properties used.** For this proof we only need: (i) all divergences appearing in $\mathcal{L}_{ign\_sm}$ are nonnegative and $Div$ satisfies identity of indiscernibles and the induced drift in Appendix A.3 is bounded with $\sup > 0$.

**ELBO maximality conditions.** We use the standard equivalence: ELBO is maximal if and only if

$$p_\theta(x) = p_{data}(x), \qquad q_\varphi(z \mid x) = p_\theta(z \mid x) \quad \text{for } x \sim p_{data}. \tag{51}$$

**Deterministic decoder at ELBO optimum.** Assume that at the ELBO optimum the decoder is deterministic (while the encoder can be arbitrary): there exists a measurable map $\mathcal{D} : \mathcal{Z} \to \mathcal{X}$ such that

$$p_\theta(x \mid z) = \delta_{\mathcal{D}(z)}(x). \tag{52}$$

Let $Z \sim p(z)$ and define $X := \mathcal{D}(Z)$. Then the induced model marginal is

$$p_\theta(x) = \mathcal{D}_\sharp p(z). \tag{53}$$

At ELBO maximum, $p_\theta = p_{data}$ by (51), so $\mathcal{D}_\sharp p(z) = p_{data}$.

**True posterior via Bayes** Under (52), the joint law of $(Z, X)$ is well-defined by $Z \sim p(z)$ and $X = \mathcal{D}(Z)$. Since $\mathcal{X}, \mathcal{Z}$ are Polish (standard Borel), there exists a regular conditional distribution (also called a disintegration)

$$p_\theta(dz \mid x) \quad \text{such that} \quad \mathbb{P}(Z \in A, X \in B) = \int_B p_\theta(A \mid x) \, p_\theta(dx) \tag{54}$$

for all measurable $A \subseteq \mathcal{Z}, B \subseteq \mathcal{X}$. This is exactly the mathematically correct meaning of $p_\theta(z \mid x) = \frac{p_\theta(x \mid z) p(z)}{p_\theta(x)}$ in the deterministic (Dirac) likelihood case.

**Lemma A.7** (Posterior is supported on the fiber). *Let $X = \mathcal{D}(Z)$ almost surely and let $p_\theta(dz \mid x)$ be any regular conditional distribution of $Z$ given $X$. Then for $x \sim p_\theta$,*

$$p_\theta\big(\{z : \mathcal{D}(z) = x\} \mid x\big) = 1. \tag{55}$$

*Proof.* Consider the event $E := \{(z, x) : \mathcal{D}(z) = x\}$. By construction $X = \mathcal{D}(Z)$ a.s., hence $\mathbb{P}(E) = 1$. By the defining property of conditional distributions,

$$1 = \mathbb{P}(E) = \int_{\mathcal{X}} p_\theta(E_x \mid x) \, p_\theta(dx), \quad \text{where } E_x := \{z : \mathcal{D}(z) = x\}. \tag{56}$$

Since the integrand is in $[0, 1]$, we must have $p_\theta(E_x \mid x) = 1$ for $x \sim p_\theta$. $\square$

**ELBO maximality pins down the encoder.** At ELBO maximum, (51) requires

$$q_\varphi(dz \mid x) = p_\theta(dz \mid x) \quad \text{for } x \sim p_\theta. \tag{57}$$

Therefore, by Lemma A.7, for $x \sim p_\theta$,

$$q_\varphi\big(\{z : \mathcal{D}(z) = x\} \mid x\big) = 1. \tag{58}$$

**Define the statistical-manifold operator used in $\mathcal{L}_{ign\_sm}$.** For any probability measure $\pi \in \mathcal{P}(\mathcal{Z})$, define the operator

$$\mathcal{T}(\pi) \triangleq \int_{\mathcal{Z}} q_\varphi(\cdot \mid \mathcal{D}(z)) \, \pi(dz) \ \in \ \mathcal{P}(\mathcal{Z}). \tag{59}$$

This is precisely the decode-encode push on the statistical manifold: sample $z \sim \pi$, decode $x = \mathcal{D}(z)$, then output the posterior measure $q_\varphi(\cdot \mid x)$ and average over $z$. In your $\mathcal{L}_{ign\_sm}$, the three terms correspond to: (i) stability: $q_\varphi(\cdot \mid x)$ vs. $\mathcal{T}(q_\varphi(\cdot \mid x))$ for $x \sim p_{data}$; (ii) tightness: $\mathcal{T}(\mathcal{T}(\delta_z))$ vs. $\mathcal{T}(\delta_z)$ for $z \sim p(z)$ (with frozen copy); (iii) idempotence: the same idempotence comparison (again with frozen copy). The exact divergence form does not matter here, only nonnegativity and $Div(\pi, \pi) = 0$.

**Lemma A.8** (Measure idempotence at ELBO optimum). *Under* (58)*, the operator $\mathcal{T}$ satisfies:*

1. *For $x \sim p_\theta$, $\mathcal{T}\big(q_\varphi(\cdot \mid x)\big) = q_\varphi(\cdot \mid x)$.*

2. *For $z \sim p(z)$, $\mathcal{T}\big(\mathcal{T}(\delta_z)\big) = \mathcal{T}(\delta_z)$.*

*Proof.* (1) Fix $x$ in the set of (58) and let $\pi = q_\varphi(\cdot \mid x)$. Then by (58), for $\pi$ a.e. $z$ we have $\mathcal{D}(z) = x$. Therefore the integrand in (59) is constant $\pi$:

$$\mathcal{T}(\pi) = \int q_\varphi(\cdot \mid \mathcal{D}(z)) \, \pi(dz) = \int q_\varphi(\cdot \mid x) \, \pi(dz) = q_\varphi(\cdot \mid x) = \pi. \tag{60}$$

(2) Fix $z_0$ and let $x_0 = \mathcal{D}(z_0)$. Then $\mathcal{T}(\delta_{z_0}) = q_\varphi(\cdot \mid x_0)$ by (59). Applying (1) with $x = x_0$ yields $\mathcal{T}(q_\varphi(\cdot \mid x_0)) = q_\varphi(\cdot \mid x_0)$. Hence $\mathcal{T}(\mathcal{T}(\delta_{z_0})) = \mathcal{T}(\delta_{z_0})$. $\square$

$\mathcal{L}_{ign\_sm} = 0$ **under the ELBO-optimal Dirac-decoder solution.** By Lemma A.8, the arguments compared by the stable/tightness/idempotence divergences inside $\mathcal{L}_{ign\_sm}$ are identical measures almost surely. Since $Div(\pi, \pi) = 0$ and $Div \geq 0$, each of the three components equals 0 and thus

$$\mathcal{L}_{ign\_sm}(\theta, \varphi) = 0 \quad \text{under} \quad (51), (52), (57). \tag{61}$$

**Conclude: VIAE minimizer must minimize VAE.** Let $\mathcal{L}^\star_{VAE} := \inf_{\theta,\varphi} \mathcal{L}_{VAE}(\theta, \varphi)$. Because $\mathcal{L}_{ign\_sm} \geq 0$,

$$\inf_{\theta,\varphi} \mathcal{L}_{VIAE} = \inf_{\theta,\varphi} \big(\mathcal{L}_{VAE} + \mathcal{L}_{ign\_sm}\big) \ \geq \ \inf_{\theta,\varphi} \mathcal{L}_{VAE} = \mathcal{L}^\star_{VAE}. \tag{62}$$

On the other hand, the ELBO-optimal Dirac-decoder construction above achieves $\mathcal{L}_{VAE} = \mathcal{L}^\star_{VAE}$ and $\mathcal{L}_{ign\_sm} = 0$ by (61), so the lower bound is tight and

$$\inf_{\theta,\varphi} \mathcal{L}_{VIAE} = \mathcal{L}^\star_{VAE}. \tag{63}$$

Therefore any global minimizer $(\theta^\dagger, \varphi^\dagger)$ of $\mathcal{L}_{VIAE}$ must satisfy $\mathcal{L}_{VAE}(\theta^\dagger, \varphi^\dagger) = \mathcal{L}^\star_{VAE}$, i.e. it is also a VAE global minimizer. In particular, at the VIAE optimum we have $p_{\theta^\dagger}(x) = p_{data}(x)$, which establishes the generative ability.

$\square$

# B. High Resolution Experiment

To evaluate the scalability and stability of our proposed framework on higher-dimensional data, we trained the Variational Idempotent Autoencoder (VIAE) on high-resolution datasets, FFHQ at $256 \times 256$ resolution.

As shown in Figure 6, the model is capable of generating consistent and high-fidelity samples with less noise. The idempotent loss plays a crucial role here; it prevents the "over-smoothing" often observed in standard VAEs when scaling up to larger images. The learned latent manifold successfully captures fine-grained details such as hair texture and facial expressions, demonstrating that the theoretical advantages of measure-theoretic idempotence hold in high-dimensional pixel space.

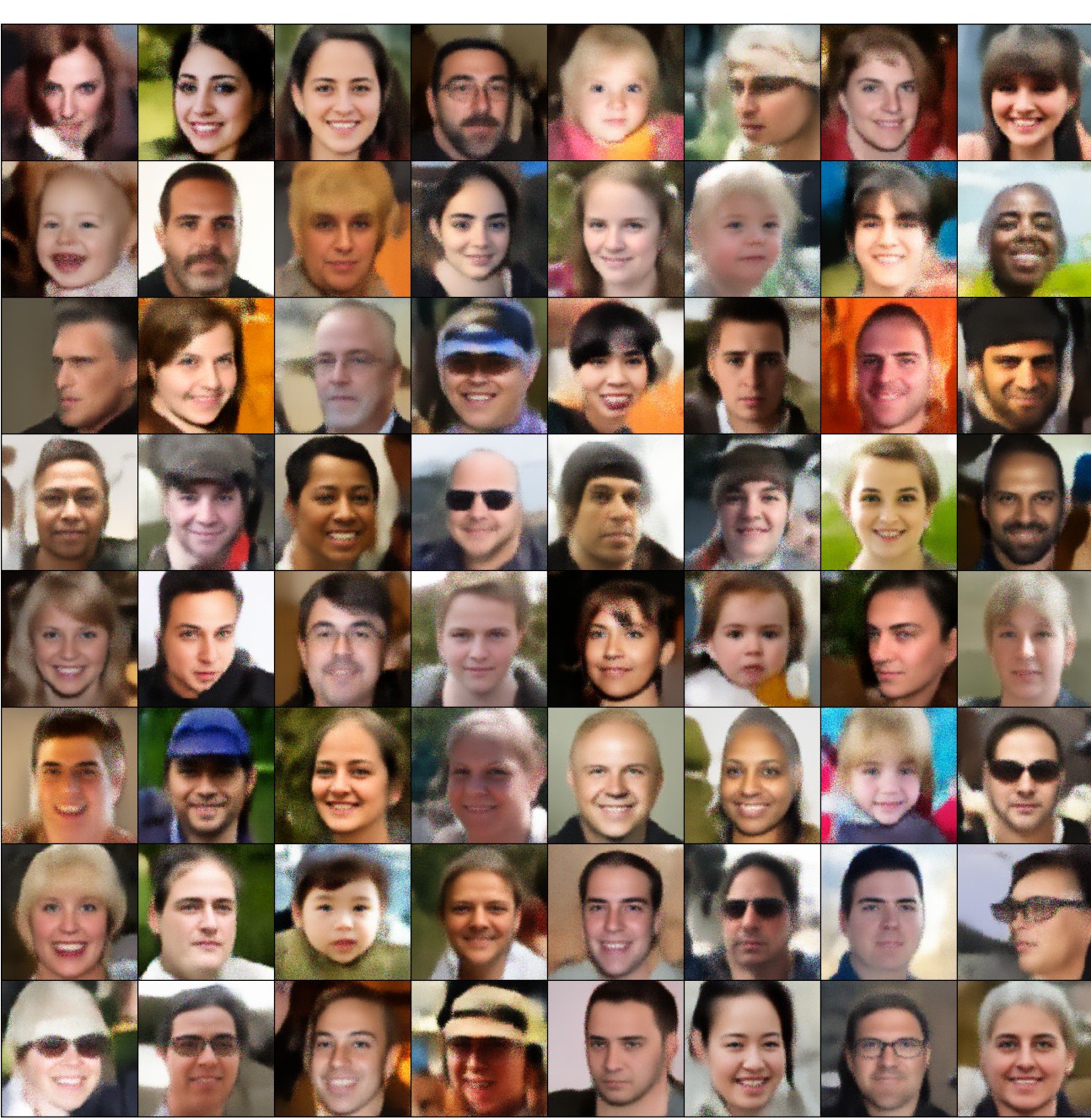

*Figure 6.* Uncurated samples generated by VIAE at $256 \times 256$ resolution. The model generates sharp and diverse images, maintaining structural coherence without the need for multi-step diffusion sampling.

