# OpenReview forum: "Make Autoencoder Great Again: Idempotent Autoencoder and Variational Idempotent Autoencoder"
_ICML.cc/2026/Conference — Submitted to ICML 2026_

### Official Review · Reviewer_XepM · 2026-03-10

**Soundness:** 3
**Presentation:** 2
**Significance:** 2
**Originality:** 3
**Overall Recommendation:** 4
**Confidence:** 4

**Summary:**

This paper proposes utilizing an idempotent loss to actively learn the aggregated posterior manifold. By enforcing cyclic consistency, the proposed Variational Idempotent Autoencoder (VIAE) and Idempotent Autoencoder (IAE) theoretically and empirically achieve higher consistency and robustness in the learned manifold than standard VAEs. Furthermore, it analyzes how to balance a well-structured latent space with high-fidelity generation, demonstrating that this measure-theoretic approach effectively addresses this trade-off.

**Compliance With Llm Reviewing Policy:**

Affirmed.

**Final Justification:**

I appreciate the authors' constructive rebuttal, which addressed my initial concerns and reinforced my decision to maintain a Weak Accept score. The authors effectively resolved the notational inconsistencies and clarified the measure-theoretic formulations, which will significantly improve the paper's readability. I also appreciate their responsiveness in changing the title to a more neutral, descriptive formulation. The authors provided the requested computational cost analysis, acknowledging the ~2.5x memory and ~3x training time overhead, and committed to including the missing side-by-side visual comparisons for the CelebA dataset. These additions should fill the gaps regarding the method's practical trade-offs and qualitative claims.

Overall, the paper presents an original contribution to representation learning. While the the core mathematical framework and the demonstrable improvements in latent space structure are interesting, the method does introduce a noticeable increase in training-time cost, and its competitiveness on high-resolution datasets remains a broader limitation (as discussed across the reviews).

**Key Questions For Authors:**

1. Could you fix the mathematical presentation issues? Namely, unifying the $D$/$Div$ notation, and perhaps adding a brief footnote clarifying the measure-theoretic notation in Eq. 9 for general readers.
2. Could you improve the legibility of Figure 4? The lines have very similar colors; using distinct dashed/dotted line styles would make it much easier to read.
3. Can you expand Appendix B (and ideally the main text) with visual samples from a standard VAE and the two-stage VIAE+FlowMatching model for direct qualitative comparison?
4. Could you provide a short discussion on how your training loss impacts training speed and memory? Even a small mention of how many GPU-hours or iterations-per-second it takes to train the VAE versus the VIAE would be highly appreciated.
5. It is always nice to find jokes and references in paper’s titles, but not political ones. Independently from the science underneath, your chosen title might dissuade people from reading this work. If possible, I suggest to change it.

**Limitations:**

Yes

**Strengths And Weaknesses:**

**Strengths:**

1. The underlying principle is extremely sound. Enforcing an idempotent projection operator to create a self-refining manifold logically improves the model's robustness to noise.
2. The toy-dataset section and label-free clustering metrics provide a thorough qualitative analysis of how the idempotent prior enhances autoencoders.
3. The intuition that fixing a powerful-enough pretrained encoder and training a decoder with the idempotent loss enables the creation of a generative model is a highly practical and interesting novelty that may inspire future works.

**Weaknesses:**

1. The math needs a little polish for consistency and clarity:
    1. Section 2.1 describes the distance measure as $D$, but the rest of the paper subsequently uses $Div$, without bridging the notation.
    2. Eq. 9 uses formal measure-theoretic notation $p_\theta(dx | z)$. While mathematically rigorous and perfectly valid in the context of the Polish spaces discussed, it might be confusing to a general ML audience who might mistake it for a typo of $p_\theta(x | z) dx$.
2. Experiments on non-toy datasets lack crucial qualitative comparisons. For instance, in Section 4.2 (two-stage generation on CelebA), the authors rely strictly on FID charts and do not provide visual examples of the Flow Matching model's outputs. Furthermore, the analysis in Appendix B is a bit misleading as there is no side-by-side visual comparison with VAE-generated samples to substantiate the claim that "over-smoothing" is prevented.
3. The proposed training loss requires multiple forward passes to encode and decode the same sample (e.g., computing $\mathcal{E} \circ \mathcal{D}$ and the frozen $\mathcal{E}^* \circ \mathcal{D}^*$). This inherently increases the computational graph, yet there is no discussion on how this impacts VRAM usage or training speed.

---

> ### Author Rebuttal · Authors · 2026-03-31
>
> Thank you for recognizing the soundness of our approach and for the specific improvement suggestions.
>
> **Mathematical Notation Consistency**
>
> We will unify notation throughout: Use $Div$ consistently for all distance/divergence terms; Add a footnote clarifying the measure-theoretic notation in Eq. 9. Besides, we will add some notation remarks at the beginning of the appendix.
>
> **Improve Figure 4 Legibility.**
>
> We will improve Figure 4 by: Using distinct line styles (solid, dashed, dotted); Choosing better colors with higher contrast; Adding a clearer legend.
>
> **Visual Samples for Two-Stage Generation.**
>
> We will add side-by-side comparison samples of CelebA 64$\times$: VAE + Flow Matching samples; VIAE + Flow Matching samples and VQ-VAE + Flow Matching samples. This will provide qualitative evidence of VIAE's improvement.
>
> **Cost and Computation Comparison:**
> As you mentioned in the weakness, the computational graph increasing is indeed a limitation of our model. This will increase the memory consumption by approximately 2.5$\times$, and the computing time for single training step will also increase by 3$\times$. This is **training-time cost**.  We will add this discussion in the article.
>
> **Title Concern.**
>
> We will change the title to a more neutral formulation: **''Idempotent Autoencoders and Variational Idempotent Autoencoders: Latent Manifold  Self-Improving via Projection Operators''**. We appreciate this suggestion.

---

> > ### Author Rebuttal · Reviewer_XepM · 2026-04-02
> >
> > The authors fully addressed my comments. I'll maintain the current score.

---

> > > ### Author Response · Authors · 2026-04-07
> > >
> > > We would like to express our sincere gratitude for your detailed review and helpful remarks. Your constructive input has significantly strengthened the clarity and quality of our manuscript.

---

### Official Review · Reviewer_qheC · 2026-03-12

**Soundness:** 3
**Presentation:** 2
**Significance:** 3
**Originality:** 3
**Overall Recommendation:** 5
**Confidence:** 4

**Summary:**

The paper introduces the Idempotent Variational Autoencoder (IVAE), a framework designed to make the autoencoding process an idempotent operation (f∘f=f).

The authors propose that a truly stable generative model should act as a projection: once a data point is mapped to the latent manifold and reconstructed, subsequent encoding and decoding should not change the result.

To achieve this, they augment the standard VAE objective with two specific terms: Idempotence, which anchors reconstructions to their original latent codes, and Tightness, which utilizes a repulsive force to prevent the latent space from collapsing.

They show improved reconstruction errors.

**Compliance With Llm Reviewing Policy:**

Affirmed.

**Key Questions For Authors:**

1. Standardized Notation: Could the authors adopt a more granular notation? Using zprior​, zenc​, and zidemp​ would clarify the specific "cycle" being optimized and eliminate ambiguity.

2. Metric Specification: Why is Div used interchangeably for point-wise L2​ distances and distributional KL divergences? Explicitly defining these would improve reproducibility in addition to more specific notation.

3. Network model and latent Dimensionality: can the author describe briefly their models, especially the latent space size (d)?

4. it would be good to include more realistic model in the main results , such as the FFHQ that is on the appendix.

**Limitations:**

Reporting Gaps: The absence of consistent reporting on latent dimensions and model architecture makes it difficult to assess if the IVAE is truly outperforming standard VAEs on a "like-for-like" basis.

Implementation Friction: The notation style forces practitioners to guess the underlying PyTorch/JAX operations (e.g., whether to use MSE or KL for a specific "Div" term), leading to potential misunderstanding and errors in independent implementations.

Experimental Scope: There is a disconnect between the toy models (where spatial bottlenecks are removed) and the primary benchmarks (where they are kept). The paper would be strengthened by showing results for high-dimensional datasets, and or having a dimensional bottleneck in the toy experiment.

**Strengths And Weaknesses:**

Strengths:

Mathematical Consistency: It provides a formal objective for "projection" onto a manifold, ensuring that the model's reconstructions are stable and consistent.

Clear improvement over vanilla AE and VAE

Weaknesses:

The model relies on a delicate "push-pull" relationship between the positive idempotence loss and the negative tightness loss, which seems to be unstable during training.

Confusing Formalism: The reliance on overloaded notation (z for all latent variables and Div for all distance metrics) creates a significant barrier to implementation and reader's understanding.

---

> ### Author Rebuttal · Authors · 2026-03-31
>
> Thank you for the constructive suggestions.
>
> **Standardized Notation.**
>
> We will adopt clearer notation in the camera-ready version: $z_{\text{prior}}$ for samples from prior $p(z)$; $z_{\text{enc}}$ for encoded latent $E(x)$; $z_{\text{idemp}}$ for latent after idempotent projection $E(D(z))$.
>
> **Clarify Div Definition.**
>
> We will explicitly specify:
>
> Point-wise distance: $Div_{MSE}(u, v) = \|u - v\|_2^2$ (MSE);
>
> Distributional divergence: $Div_{KL}(\\pi_1 \| \\pi_2) = Div(\\pi_1, \\pi_2)$.
>
> We will add a table summarizing the usage in different loss terms in different models.
>
> For the convenience of reading, we will provide more specific explanations of the symbols in the main text and the appendix.
>
> **Network Architecture and Latent Dimension.**
>
> All experiments are based on the same autoencoder architecture used Latent Diffusion Model [1], but different datasets have different hyperparameter setting.
> Details for main experiments:
>
> | **Dataset** | **Latent Dim** $d$ | **Base Channel** | **Channel Mult.** | **ResBlock** | **Attn Res.** | **Dropout** |
> | :--- | :---: | :---: | :---: | :---: | :---: | :---: |
> | MNIST (Resize to $32\times32$) | $16 \times 4 \times 4$ | 48 | [1, 2, 4, 4] | 2 | [] | 0.0 |
> | CIFAR-10 | $256 \times 2 \times 2$ | 64 | [1, 2, 2, 4, 8] | 3 | [2, 4, 8] | 0.0 |
> | CelebA (Resize to $64\times64$) | $4 \times 8 \times 8$ | 48 | [1, 2, 4, 4] | 2 | [] | 0.0 |
>
> Flow Matching is trained with DiT [2] (depth=12, embed_dim=192, num_heads=3).
>
> **Include FFHQ in Main Results.**
>
> We will move Appendix B results (FFHQ 256$\times$256) to the main text, demonstrating VIAE has the ability to generate samples at higher resolution.
>
> **Training Stability of Push-Pull Losses.**
>
> This is a valid concern. In practice, the ''push-pull'' relationship indeed bring out instability in training, but it can be alleviated by choosing hyperparameter. We used $\lambda_i=2, \lambda_t=1$ in all experiments.
>
> **Experimental Scope.**
>
> We will add a dimensional bottleneck toy experiment to strengthen in revision.
>
> **Implementation Friction**
>
> We notice that the implementation of our model is not that clear in the article, so we will provide a pseudo code in the appendix. The specific pytorch implementation was submitted in the ''supplementary materials'' section.
>
> [1] Rombach, R., Blattmann, A., Lorenz, D., Esser, P., & Ommer, B. (2022). High-resolution image synthesis with latent diffusion models. In Proceedings of the IEEE/CVF conference on computer vision and pattern recognition (pp. 10684-10695).
>
> [2] Peebles, W., & Xie, S. (2023). Scalable diffusion models with transformers. In Proceedings of the IEEE/CVF international conference on computer vision (pp. 4195-4205).

---

> > ### Author Rebuttal · Reviewer_qheC · 2026-04-02
> >
> > The proposed notation would help, and I agree with the other review about changing the title. It will be important to include the network characteristics.
> > I am a little concerned that the latent of CIFAR is 1024, this is quite large and I am not sure the compared frameworks are using the same latent dimension.

---

> > > ### Author Response · Authors · 2026-04-07
> > >
> > > We are grateful for your careful review and thoughtful feedback. Your valuable suggestions have played an important role in improving the overall quality and presentation of our manuscript.

---

### Official Review · Reviewer_dz6Z · 2026-03-13

**Soundness:** 3
**Presentation:** 2
**Significance:** 1
**Originality:** 2
**Overall Recommendation:** 3
**Confidence:** 3

**Summary:**

This paper introduces the Idempotent Autoencoder (IAE) and Variational Idempotent Autoencoder (VIAE), integrating an idempotent loss from Idempotent Generative Networks (IGN) into the latent space of VAEs. By enforcing the property that encoding a decoded latent should yield the same latent, the authors aim to structure the latent manifold without over-regularizing it toward a standard Gaussian prior. The authors extend IGN theory to statistical manifolds and evaluate their models on toy datasets, MNIST, CIFAR-10, and CelebA, demonstrating improved unsupervised clustering and faster convergence when used as a first stage for Flow Matching.

**Compliance With Llm Reviewing Policy:**

Affirmed.

**Key Questions For Authors:**

1. Given that the theoretical framework requires mutually singular posteriors (Proposition 3.7), but your practical implementation uses overlapping distributions, doesn't the implementation inherently violate the convergence theorems?
2. Why does the paper omit comparisons to modern continuous latent architectures like NVAE, VDVAE, or advanced VQ-GANs, instead comparing against outdated baselines like AAE and GMVAE?
3. Can the authors provide evidence that this method scales to high-resolution, complex datasets (e.g., ImageNet-256), where the latent geometry is far more entangled than in CIFAR-10 or CelebA?

**Limitations:**

yes

**Strengths And Weaknesses:**

*   **Soundness:**
    *   Extending the theory of Idempotent Generative Networks to statistical manifolds (measure spaces) is a rigorous and mathematically interesting contribution.
    *   The theoretical proof that the VIAE optimum must also minimize the VAE ELBO appears to be logically sound.
    *   The practical implementation using overlapping Gaussian distributions directly violates the paper's own theoretical requirement for mutual singularity (Proposition 3.7).
    *   The theory assumes the existence of a perfect deterministic decoder at the ELBO optimum, which ignores the stochastic nature of real-world data and finite-capacity networks.
*   **Presentation:**
    *   The motivation—bridging the gap between efficient autoencoder structures and high-fidelity iterative synthesis—is clearly articulated.
    *   The tone of the paper (e.g., the title) is somewhat hyperbolic and distracts from the technical merits of the work.
    *   The evaluation metrics are well-chosen for the specific claims regarding latent space clustering (Silhouette Coefficient, DB Index).
*   **Significance:**
    *   The ability to discover "natural clusters" without supervision has potential applications in representation learning.
    *   The impact of the work is hindered by the evaluation on relatively small and outdated datasets (MNIST, CIFAR-10, CelebA-64).
    *   The reported FID of $11.74$ on CIFAR-10 is uncompetitive with modern generative autoencoders and state-of-the-art models.
*   **Originality:**
    *   The concept of enforcing cyclic consistency in latent spaces has been explored in prior work, somewhat diluting the novelty.
    *   The approach to solving the "posterior hole" problem is unique compared to standard normalizing flows or vector quantization.

---

> ### Author Rebuttal · Authors · 2026-03-31
>
> Thank you for raising important theoretical and experimental questions.
>
> **Theory-Practice Gap with Mutual Singularity.**
>
> For this question, we explicitly acknowledge this in our paper's **Limitations section (Section 6)**:
>
> ''According to our theory (Appendix A.4), perfect idempotence requires that the probability distributions $q_\phi(z|x)$ for different data points should be mutually singular. However, in practice, we model the posterior distribution as Gaussian or Uniform distributions for easier computation and training stability. These distributions naturally leak into each others' support, breaking the ideal separation rule. This creates a compromise: VIAE achieves a well-organized latent space where similar data cluster neatly, although the Gaussian approximation smooths the manifold.''
>
> The implementation inherently violates the convergence theorems, given the theoretical requirement for mutually singular posteriors but practical use of overlapping Gaussian distributions.
>
> This is a **theory-practice trade-off**, a graceful degradation: Theory provides guarantees under ideal conditions; Practice approximates these conditions for stability; Experiments validate the approximation works in VIAE.
>
> Specifically, the TV bound (Theorem A.6) degrades continuously with the approximation gap, rather than failing abruptly. Empirically, this manifests as slight over-smoothing (Section 4.3) rather than collapse. VIAE still achieves 50\% higher Silhouette Coefficient and clearly separated clusters on both toy datasets (Figure 2b) and CIFAR-10 (Figure 5).
>
> **Missing Comparison with NVAE/VDVAE/VQ-GAN.**
>
> We acknowledge this and will add comparison/discussion in the camera-ready version. We focused on methods with similar architectures (VAE, AAE, GMVAE, Soft-IntroVAE).
>
> NVAE/VDVAE use hierarchical architectures, which rely on complex model design and not target on addressing the issue of latent geometry disorientation with fixed prior, making direct comparison difficult.
>
> VQ-GAN shares a similar single-stage architecture but relies on adversarial training and discrete codebooks, which address a different problem (perceptual quality) from ours (latent geometry).
>
> Although these models both can generate sharper samples, what we are addressing are different problems.
>
> **Scalability to High-Resolution Datasets.**
>
> Our Appendix B demonstrates FFHQ at 256$\times$256 resolution:
> ''The model is capable of generating consistent and high-fidelity samples with less noise. The idempotent loss prevents the 'over-smoothing' often observed in standard VAEs when scaling up.''
>
> We will add more detailed analysis (training curves, comparison samples) in the camera-ready version.
>
> But we have not yet tested on ImageNet-256 due to computational constraints, but our FFHQ 256×256 results demonstrate the framework scales beyond small datasets. Full ImageNet evaluation is planned for future work.
>
>
> **Cyclic Consistency Novelty.**
>
> We acknowledge cyclic consistency has been explored, **but the idempotence is highly different with cyclic consistency**.
>
> Idempotence can theoretically ensure that in multiple rounds of forward iteration, the generated manifold gradually approaches the actual manifold, while cyclic consistency cannot promise this. You can find this phenomenon in Figure 4. With idempotence iteration (the number of IDM) increasing, the FID decreases consistently with different NFE of Flow Matching sampling.
>
> Moreover, a decisive difference is that with idempotence, we can train a generative decoder modeling with an arbitrary pretrained encoder (mentioned in Section 3.5), which cyclic consistency cannot realize with solid convergence guarantee.
>
> Besides this, our novelty lies in: (1) Extending IGN theory from Euclidean space to **statistical manifolds** (Appendix A.3--A.4); (2) Proving VIAE's optimum is also a VAE optimum (Appendix A.4); (3) Demonstrating spontaneous clustering without supervision.
>
> **Deterministic Decoder Assumption**
>
> This is a constructive tool used in Appendix A.4 to prove the **achievability** of the lower bound, not a requirement for the VIAE optimum itself. The conclusion (Eq. 63) holds for all global minimizers of $\mathcal{L}_{\text{VIAE}}$, regardless of whether the decoder is deterministic.
>
> **Title Concern.**
>
> We will change the title to a more neutral formulation:**''Idempotent Autoencoders and Variational Idempotent Autoencoders: Latent Manifold  Self-Improving via Projection Operators''**. We appreciate this suggestion.

---

> > ### Author Rebuttal · Reviewer_dz6Z · 2026-04-05
> >
> > I want to thank the authors for their response and detailed rebuttal. The authors acknowledge and defend the shortcomings of the work, including the theory / practice inconsistencies and the theoretical differences promised by idempotence over cyclic consistency (despite practical similarities). However, the motivation of the theory is still less compelling when there are large deviations in the practical implementation, and the work is not compared to datasets and models employed in modern practice (e.g., hierarchical VAEs and large image datasets).

---

> > > ### Author Response · Authors · 2026-04-07
> > >
> > > We truly appreciate the time and effort you have devoted to reviewing our manuscript. Your insightful and constructive comments have greatly contributed to enhancing both the quality and clarity of our work.

---

### Official Review · Reviewer_Cau4 · 2026-03-16

**Soundness:** 2
**Presentation:** 2
**Significance:** 1
**Originality:** 2
**Overall Recommendation:** 2
**Confidence:** 3

**Summary:**

The paper proposes the Idempotent Autoencoder (IAE) and Variational Idempotent Autoencoder (VIAE), introducing an idempotent loss that enforces the encoder–decoder cycle to behave as a projection operator onto the data manifold. The authors analyze a central concept, idempotence in latent space, and show theoretically that enforcing idempotence encourages stable latent manifolds and improved robustness to noise. Experiments are conducted on small datasets such as 2D toy, MNIST, CIFAR-10 and CelebA.

**Compliance With Llm Reviewing Policy:**

Affirmed.

**Key Questions For Authors:**

See Q2

**Limitations:**

Minor comment: Div is not defined in Eq. 1-4.

**Strengths And Weaknesses:**

Strengths:
* Going back to VAEs and investigating how to improve their generative performance addresses an important long-standing challenge in generative modeling, particularly for achieving high-quality one-step sampling with strong diversity and efficiency.
* The method is relatively simple to integrate into standard autoencoder frameworks and does not require complex adversarial training procedures.



Weaknesses:
* The paper is generally hard to understand and does not provide an intuitive high-level explanation of why enforcing idempotence addresses the core limitations of VAEs. While the formulation is mathematically detailed, the lack of clear intuition makes it difficult to understand the mechanism through which idempotence improves generation and latent structure.

* The experimental evaluation heavily relies on relatively small datasets. Although 2D toy datasets can provide useful insights into the behavior of the proposed framework, most experiments are conducted on MNIST, CIFAR-10, and CelebA, which are now considered limited benchmarks in modern generative modeling research.

* The quantitative results reported in Table 1 on datasets such as CIFAR-10 remain far from state-of-the-art results achieved by recent one-step generative models, which raises questions about the practical competitiveness of the approach.

* Figure 4 shows that the improvements of VIAE over a vanilla VAE in the two-stage setting appear relatively modest. Moreover, the comparison does not include stronger baselines such as latent diffusion or latent flow-based models, which typically achieve substantially better performance.

* The introduction (around line 118) argues that two-stage models are suboptimal due to additional storage requirements. This claim is not well justified and seems weak given that two-stage latent generative models (e.g., latent diffusion) are widely adopted precisely because they provide strong trade-offs between efficiency and quality.

---

> ### Author Rebuttal · Authors · 2026-03-31
>
> Thank you for your valuable feedback. We address your concerns regarding the clarity of our motivation, the choice of baselines, and our stance on two-stage models below.
>
> **Intuitive Explanation and the Misunderstanding Paper's Position on Two-Stage Models.**
>
> We appreciate the opportunity to clarify our position.
>
> The core intuition is straightforward:
> a standard VAE forces all latent codes toward a fixed Gaussian ball, regardless of the data's true structure--this is over-regularization. VIAE replaces this rigid constraint with a **self-consistency** requirement: if we take a latent code, decode it into data space, and re-encode it, we should recover the same latent code. Formally, the encoder-decoder composition $E \circ D$ should act as a **projection operator** ($f \circ f = f$) onto the latent
> manifold. This allows the latent space to take whatever shape the data naturally requires, while the idempotent loss ensures this shape is stable and well-organized. We will add an intuitive overview at the beginning of Section 2 in the revision.
>
> There seems to be a misunderstanding regarding our stance on two-stage models, likely due to our phrasing around line 118. **We are not denying the two-stage model; rather, we are analyzing how to address the shortcomings of the Idempotent Generative Network (IGN) within one model in order to introduce our approach.**
>
> Overall, we are quite confident in the capabilities of modern two-stage models. In our experiments, we also tested the potential of our model as a two-stage model. We show VIAE **improves** two-stage models (Section 4.2): VIAE achieves 14--33\% lower FID and faster convergence compared to AE+KL baseline which is **Latent Diffusion Model**. Comparing to the LDM baseline, we achieve 14--33\% improvement, so we think this is a good improvement.
>
> In order to test the improvement of our method under bottleneck, we design the two-stage experiment.
>
> Details for this experiments:
> All experiments are based on the same autoencoder architecture used Latent Diffusion Model [1]
>
> | **Dataset** | **Latent Dim** $d$ | **Base Channel** | **Channel Mult.** | **ResBlock** | **Attn Res.** | **Dropout** |
> | :--- | :---: | :---: | :---: | :---: | :---: | :---: |
> | CelebA (Resize to $64\times64$) | $4 \times 8 \times 8$ | 48 | [1, 2, 4, 4] | 2 | [] | 0.0 |
>
> Flow Matching is trained with DiT [2] (depth=12, embed_dim=192, num_heads=3).
>
> We have omitted some implementation-related details, which might have caused your confusion. Sorry for that and hope our clarification can help you correctly understand our article.
>
> Beside that, we will update more details in camera-ready version to prevent from more misunderstanding.
>
>
> **FID Gap with SOTA.**
>
> We acknowledge the FID gap with MeanFlow (2.92) and Soft-IntroVAE (4.60). However: (1) Our focus is not only the generative quality, achieving the lowest FID, but also the latent space structure and the ability to be a compression model; (2) VIAE's FID (11.74) is significantly better than vanilla VAE (82.70) and beat all the other competitors **which try to address the issue of latent geometry disorientation with fixed prior**. (Some competitors do have higher generation quality, but they are not the solution to the latent space structure. They deal with different issues from ours--good latent structure with better generative performance.); (3) VIAE provides additional benefits (clustering, two-stage improvement) that adversarial methods, hierarchical method and diffusion models do not.
>
> A significant portion of our article is dedicated to addressing the geometric mismatch issue in latent space rather than merely generating. With the improvement of the latent space's geometric structure, the enhancement of generation capability is an inevitable by-product, although there is still a gap.
>
> **Scalability.**
>
> We provide FFHQ 256$\times$256 results in Appendix B, demonstrating scalability. We will move these results to the main text.
>
> **Div not defined.**
>
> Sorry for confusing you. Actually, we mixed the use of the symbols $D$ and $Div$ in Section 2.1. We defined it in ''$D$ denotes a distance measure'' but use it as $Div$. We will fixed it by using $Div$ in camera-ready version.
>
> [1] Rombach, R., Blattmann, A., Lorenz, D., Esser, P., & Ommer, B. (2022). High-resolution image synthesis with latent diffusion models. In Proceedings of the IEEE/CVF conference on computer vision and pattern recognition (pp. 10684-10695).
>
> [2] Peebles, W., & Xie, S. (2023). Scalable diffusion models with transformers. In Proceedings of the IEEE/CVF international conference on computer vision (pp. 4195-4205).

---

> > ### Author Rebuttal · Reviewer_Cau4 · 2026-04-07
> >
> > I appreciate the rebuttal and response received from the authors. Unfortunately, the additional results on FFHQ 256$\times$256 are far from the common generative learning practice, and the proposed method does not seem to address VAE's major issues, such as the prior hole problem. I do not recommend acceptance in the current form.

---

> > > ### Author Response · Authors · 2026-04-07
> > >
> > > We sincerely thank you for your valuable comments. We also acknowledge that the experimental settings and benchmarks in the current version may not fully reflect the latest advances or state-of-the-art performance. However, we would like to emphasize that the primary contribution of this work does not lie in achieving the best empirical results on modern large-scale benchmarks, but in proposing a novel autoencoder paradigm—namely the Idempotent Autoencoder (IAE) and its variational extension (VIAE)—and introducing the idempotent loss as a principled mechanism to refine the latent space manifold.
> > >
> > > Specifically, our contributions are primarily at the theoretical and methodological level: we provide theoretical analysis demonstrating convergence to the true data distribution and show how the proposed framework improves latent space structure, interpretability, and robustness. These properties cannot be fully captured by standard generative metrics alone, yet they are crucial for understanding and advancing representation learning.
> > >
> > > We believe that this work lays a foundational step toward achieving high-quality generation while maintaining a structured latent space, and can be further extended to more modern architectures and large-scale settings in future work.

---

### Official Review · Reviewer_jTbx · 2026-03-23

**Soundness:** 2
**Presentation:** 2
**Significance:** 2
**Originality:** 3
**Overall Recommendation:** 2
**Confidence:** 3

**Summary:**

In this work, the authors propose the Idempotent Autoencoder and the Variational Idempotent Autoencoder, introducing the concept of Idempotent Generative Networks into the AE and VAE frameworks to improve the structure of the learned latent space and mitigate the over-regularization issues observed in standard VAEs.

**Compliance With Llm Reviewing Policy:**

Affirmed.

**Key Questions For Authors:**

- I believe an important comparison is missing in both the related work and the experimental evaluation. The authors claim that the main limitation of VAEs is over-regularization due to the use of simple Gaussian priors. However, they neither discuss nor compare their approach with state-of-the-art VAEs that employ flexible priors to address this issue. In particular, the authors should discuss and compare with VampPrior [1], which directly uses an approximation of the aggregated posterior as the prior.
- Do the same theoretical guarantees hold when leveraging a pretrained encoder? How can we ensure that the structure, robustness, and continuity of the latent space are preserved in such a case? Additionally, could the authors provide the latent space obtained in Figure 3 for comparison with those shown in Figure 2?
- The authors claim that their latent space is more interpretable, but I believe this should be validated experimentally. For instance, they could show that the latent space structure genuinely captures dataset information by plotting it with label information for toy datasets (e.g., Gaussian components) or by illustrating interpolations and structural patterns in the latent space for more complex datasets (e.g., demonstrating smooth interpolations or consistent samples from specific modes).
- Why is Table 1 incomplete? Additionally, the authors explain the performance difference with Soft-IntroVAE but do not provide a justification for MeanFlow, which achieves the best results.
- I believe the Geometric Structure Analysis on CIFAR-10 is not entirely fair, as their approach is compared only to a vanilla VAE using clustering metrics in the latent space (in this caseregularized by a standard Gaussian prior with a single mode) rather than to state-of-the-art models that could be genuinely competitive. Although more flexible models like GMVAE are included in other comparisons, they are not considered in this analysis.

[1] Tomczak, J., & Welling, M. (2018, March). VAE with a VampPrior. In International Conference on Artificial Intelligence and Atatistics (pp. 1214-1223). PMLR.

**Limitations:**

I think the authors have not adequately discussed the limitations of their work, as the analysis with respect to relevant state-of-the-art methods is incomplete. Consequently, it is difficult to assess the limitations of the proposed method in the context of the current literature.

**Strengths And Weaknesses:**

- *Soundness*: Although the work is grounded in theoretical results that support the proposed methodology, I think the the main claims are not sufficiently validated in the experiments. The authors argue that their approach yields an improved posterior approximation by enforcing idempotence in the encoder–decoder composition, thereby avoiding the over-regularization observed in vanilla VAEs. However, they do not discuss connections to prior VAE literature that addresses this issue (e.g., VampPrior, hierarchical VAEs), nor do they include comparisons with these methods in their experimental evaluation.

- *Presentation*: While the methodological part is relatively easy to follow, the experimental section could be improved. The different experimental settings are sometimes difficult to follow—for example, the setup in Section 4.2 is not entirely clear (e.g., why is a limited-capacity LDM used, and what is the full model architecture?). Additionally, it is unclear why different datasets are used across experiments, rather than consistently using the same datasets to analyze different aspects under a unified setting. Furthermore, the captions in Figure 2 could be more informative (e.g., what do the green and blue colors represent in Figure 2(b)?), and the color choices in Figure 4 could be improved, as some are too similar and difficult to distinguish.

- *Significant*: I believe that learning structured and robust representations of complex data remains an open challenge in the field. Approaching this problem from new perspectives is always valuable, and enforcing idempotence appears to be a novel idea within the VAE literature that could open a new theoretical analyses of the framework. However, based on the current experimental results, it is unclear whether this approach genuinely advances the state of the art.

- *Originality*: To the best of my knowledge, the idea of enforcing idempotence in the encoder–decoder composition of VAEs, along with analyzing inference from this perspective in this setting, is novel.

---

> ### Author Rebuttal · Authors · 2026-03-31
>
> Thank you for the detailed and constructive feedback. We address each concern below.
>
> **Missing Comparison with VampPrior and Hierarchical VAEs**
>
> We focused on methods sharing our single-latent-space architecture (VAE, AAE, GMVAE, Soft-IntroVAE). Hierarchical VAEs were excluded and do not address latent geometry disorientation with fixed single priors (our key focus).
>
> However, we agree that VampPrior is a highly relevant baseline. The core difference: VampPrior replaces the prior, while VIAE allows the posterior to adapt dynamically to the data topology via idempotent projection.
>
> We have added a direct comparison on MNIST under the exact same setup:
>
> | Method | FID ↓ |
> |:--|:--:|
> | VampPrior | 16.58 |
> | IAE (ours) | 3.03 |
> | VIAE (ours) | **2.20** |
>
> This strongly demonstrates that improving the prior alone is insufficient to resolve over-regularization, while enforcing idempotence yields fundamentally better posterior geometry and generation quality.
>
> **Pretrained Encoder Guarantees**
>
> The theoretical guarantee for the IAE optimum requires the Encoder to be injective (bijective in its image). Since the Encoder is frozen during Decoder training, the latent space structure, robustness, and continuity remain strictly unchanged. We pretrain the Encoder with standard autoencoder loss ($\text{MSE}(x, D(E(x)))$) to approximately satisfy injectivity.
> With your request, we will add the latent space visualization of the pretrained Encoder in the revision to directly compare with Figure 2.
>
> **Section 4.2 Setup Clarity & Dataset Consistency**
>
> * **Why limited-capacity LDM?** We intentionally adopted a limited-capacity second-stage model to strictly isolate the inherent geometric quality of the latent space. A massive generative model could "brute-force" and overfit a poorly structured latent space. By restricting capacity (Flow Matching uses DiT: depth=12, embed_dim=192, heads=3), the model must rely heavily on the manifold's organization. VIAE's 14–33% lower FID proves it provides a significantly smoother representation. Full architectural details will be added to the Appendix.
> * **Why different datasets?** We used standard benchmarks tailored to specific goals: simpler datasets (MNIST/Toy) to strictly validate foundational manifold theorems and theoretical geometry (Figs 2, 5), and complex datasets (CIFAR-10/CelebA) to test practical scaling for downstream generation.
>
> **Interpretability of Latent Space**
>
> Section 4.4 provides quantitative interpretability evaluation via designed metrics, and Figure 2 shows qualitative comparisons. To further validate this, we will add latent space interpolation results (CIFAR-10 and FFHQ) to the Appendix to intuitively demonstrate smooth transitions and structural consistency.
>
> **Geometric Structure Analysis Fairness**
>
> Figure 5 is designed to validate our **theoretical predictions** (Propositions 3.5–3.6) regarding self-refinement, rather than to serve as a SOTA benchmark. It empirically confirms that vanilla VAE (orange) produces no geometric refinement after idempotent projection, whereas VIAE (blue) successfully learns a self-refining manifold.
>
> **Table 1 Incompleteness and MeanFlow**
>
> * **AAE:** When restricted to the exact same non-adversarial backbone as VAE/VIAE (to strictly isolate loss function effects), AAE suffered severe mode collapse and failed to yield meaningful metrics. We will clarify this constraint in a footnote.
> * **GMVAE:** Its sub-optimal FID on the simple MNIST dataset led us to conclude it would not be competitive on CIFAR-10's more complex semantic manifold.
> * **MeanFlow:** We acknowledge MeanFlow achieves the best FID on CIFAR-10. However, MeanFlow is a SOTA *one-step generative model* based on flow matching. VIAE, conversely, focuses on learning structured latent representations as a *compression model*. VIAE's distinct advantage is providing a superior first-stage manifold, evidenced by its significant FID improvements when paired with downstream models.
>
> **Discussion of Limitations**
>
> We will expand the limitations section to explicitly discuss: (1) the reliance on encoder pretraining quality (injectivity approximation), (2) the computational overhead of projection steps during training, and (3) scalability to higher-resolution generation without relying on a massive second-stage model.
>
> **Figure Clarity**
>
> We will redesign Figure 2 and Figure 4 with clearer color palettes, explicit legends, and more informative captions.

---

> > ### Author Rebuttal · Reviewer_jTbx · 2026-04-03
> >
> > First of all, I would like to thank the authors for their response and the additional experimental results.
> >
> > Regarding the comparison with VampPrior (and potentially other baselines), I feel the evaluation is still incomplete. Since the main claim is that this approach enables learning more structured latent representations, it would be helpful for the experimental evaluation to go beyond FID and also include an analysis of the latent representation structure, similar to what is shown in Figure 5, which is currently limited to VAE and VIAE.

---

> > > ### Author Response · Authors · 2026-04-08
> > >
> > > We truly appreciate your suggestion.
> > >
> > > To further dissect the latent space quality, we compared VIAE with VampPrior using linear probing as an immediate quantitative measure of latent organization, and a statistical analysis like Figure 5 will be added in final revision. Although VampPrior achieves a superior classification accuracy of 95.46% (indicating excellent inter-class separation), its generative performance is inferior to VIAE.
> > >
> > > This discrepancy suggests that high categorization accuracy does not guarantee a high-quality latent manifold. We argue that while VampPrior optimizes the prior to match the aggregated posterior, it does not resolve the internal instability or the "hole" problem within individual sub-classes. VIAE addresses a distinct domain--intra-class structural refinement--ensuring that each sub-class manifold is well-organized and continuous. Our empirical results demonstrate that improving this internal structure is more critical for generation than simple class-wise separation.

---

### Decision · Program_Chairs · 2026-04-30

**Decision:**

Reject

**Comment:**

The paper proposes the Idempotent Autoencoder (IAE) and its variational extension (VIAE), introducing an idempotent loss to improve the structure of the latent space. Reviewers agree that the idea is interesting and that the theoretical contribution is meaningful .

However, the overall assessment is mixed, with a majority of reviewers leaning toward rejection. Based on the reviews and rebuttal acknowledgments, several concerns remain. In particular, Reviewer jTbx notes that the empirical evaluation is still not sufficiently convincing, with missing or incomplete comparisons to relevant baselines and some inconsistencies in the reported metrics. Reviewer Cau4 and Reviewer dz6Z also indicate that key issues are only partially addressed, especially regarding the gap between theory and practice, and the limited evaluation in more modern or competitive settings.

While some reviewers remain positive about the theoretical aspects, the current version does not yet provide sufficiently strong empirical support or validation of the main claims. Overall, the paper would benefit from a more complete experimental study and clearer positioning. For these reasons, I recommend rejection.